



# An updated estimate of radium 228 fluxes toward the ocean : how well does it constrain the submarine groundwater discharge ?

Guillaume Le Gland[1], Laurent Mémery[1], Olivier Aumont[2], and Laure Resplandy[3]

[1]LEMAR, Institut Universitaire Européen de la Mer, Plouzané, France
[2]LOCEAN, Institut Pierre Simon Laplace, Paris, France
[3]Scripps Institution of Oceanography, University of California San Diego, La Jolla, CA, USA

*Correspondence to:* Guillaume Le Gland (guillaume.legland@univ-brest.fr)

**Abstract.** Radium 228 ($^{228}$Ra), an almost conservative trace isotope of the ocean, supplied from the continental shelves and removed by a known radioactive decay ($T_{1/2} = 5.75$ yr), can be used as a proxy to constrain shelf fluxes of other trace elements, such as nutrients, iron, or rare earth elements. In this study, we perform inverse modeling of a global $^{228}$Ra dataset (including GEOSECS, TTO and GEOTRACES programs, and, for the first time, data from the Arctic and around the Kerguelen islands) to compute the total $^{228}$Ra fluxes toward the ocean, using the ocean circulation obtained from the NEMO 3.6 model with a 2° resolution. We optimized the inverse calculation (source regions, cost function) and find a global estimate of the $^{228}$Ra fluxes of $8.01 - 8.49 \times 10^{23}$ atoms yr$^{-1}$, lower and more precise than previous estimates. The largest fluxes are in the western North Atlantic, the western Pacific and the Indian Ocean, with roughly two thirds in the Indo-Pacific basin. A first estimate in the Arctic Ocean is assessed ($0.20 - 0.50 \times 10^{23}$ atoms yr$^{-1}$). Local misfits between model and data in the Arctic, the Gulf Stream and the Kuroshio regions could result from flaws of the ocean circulation in these regions (resolution, atmospheric forcing). As radium is enriched in groundwater, a large part of the $^{228}$Ra shelf sources comes from submarine groundwater discharge (SGD), a major but poorly known pathway for terrestrial mineral elements, including nutrients, to the ocean. In contrast to the $^{228}$Ra budget, the global estimate of SGD is rather unconstrained, between 1.3 and $14.7 \times 10^{13}$ m$^3$ yr$^{-1}$, due to high uncertainties on the other sources of $^{228}$Ra, especially diffusion from continental shelf sediments. Better precision on SGD cannot be reached by inverse modeling until a proper way to separate the contributions of SGD and diffusion at a global scale is found.

## 1 Introduction

Trace Elements and Isotopes (TEI) are low concentration components of the ocean, but they contain decisive information for our understanding of its dynamics. The international program GEOTRACES has been designed to improve our knowledge on the TEIs concentrations and the oceanic processes controlling their distribution, by means of observations, modeling and laboratory experiments. Since 2006, GEOTRACES cruises have been mapping the global distribution of tens of these TEIs. Some of them are studied because they constitute micronutrients for living organisms, like iron (Fe), or pollutants, like lead (Pb) and cadmium (Cd). Others are proxies of ocean dynamics or of biogeochemical processes: For instance, neodymium (Nd) is a proxy of the exchanges between seabed and seawater (Jeandel et al., 2007), and thorium 234 ($^{234}$Th) is related to the biological carbon pump (Clegg and Whitfield, 1991; Buesseler et al., 1992; Henson et al., 2011). Radium isotopes are of





particular interest as they are proxies of all TEI fluxes from sediments and continents toward the ocean. More specifically, radium isotopes have been used to estimate a still poorly known pathway to the ocean : Submarine Groundwater Discharge (SGD).

SGD is defined as the flux of water from coastal aquifers to the ocean, regardless of its composition and origin. Part of it is
meteoric freshwater, but the largest part is infiltrated seawater (Burnett et al., 2006; Moore, 2010b). In these aquifers, water gets enriched in nutrients and trace elements, released from soils and rocks or coming from land pollution, before flowing back to the ocean. It has traditionally been considered that the main coastal sources of terrestrial mineral elements to the ocean are rivers, but there is growing evidence that SGD is in fact a source of nutrients of the same order of magnitude, affecting the biogeochemistry at all scales, from coastal regions (Hwang et al., 2005; Kim et al., 2005) to ocean basins (Moore et al., 2008;
Rodellas et al., 2015). SGD is a suspected cause of algal blooms, including harmful algal blooms (LaRoche et al., 1997), and a pathway for contamination. In spite of this, their contribution is still much less precisely known than the river inputs. Because of the strong heterogeneity in their distribution and intensity, properly estimating SGD by direct methods requires an intense sampling, which is far from being fulfilled (Burnett et al., 2006; Moore, 2010a). Indirect methods should then be used: Radium (Ra) isotopes offer a great potential due to their relation to SGD and their simple chemistry.
All the four natural radium isotopes, $^{223}$Ra ($T_{1/2} = 11.4$ d), $^{224}$Ra ($T_{1/2} = 3.6$ d), $^{226}$Ra ($T_{1/2} = 1602$ yr) and $^{228}$Ra ($T_{1/2} = 5.75$ yr), are produced within the rocks by the radioactive decay of thorium. Since radium is far more soluble in water than thorium, its main source is not the decay of dissolved thorium in the ocean, but dissolution from lithogenic material. Therefore it is used as a tracer of boundary fluxes. This element is released into the ocean by three main sources located on the continental shelf: dissolution from riverine particles, diffusion from seabed sediments, and SGD, which are highly enriched. Dust inputs
account for less than 1% of all inputs (Moore et al., 2008). In the ocean, Ra is almost conservative. It is removed by radioactive decay and scavenging. Scavenging is associated with a residence time of approximately $500$ yr (Moore and Dymond, 1991), making it negligible for all Ra isotopes except $^{226}$Ra. Then the distributions of the other three isotopes, whose radioactive sinks are known, depend only on the source distribution and transport by the ocean circulation. Since their half-life time scales are small relative to the time scale of basin-wide horizontal mixing, typically a few years to a few decades, $^{223}$Ra and $^{224}$Ra are
unsuitable for large scale studies. $^{228}$Ra, whose half-life of $5.75$ yr is short enough to neglect scavenging but long enough to consider the global ocean, is suitable for global scale analyses and is thus considered here.

A simple way to use the information provided by this isotope is to make an observation-based inventory of the ocean $^{228}$Ra. At steady state, the supply of $^{228}$Ra must balance the loss from disintegration, i.e. 12% every year. According to Charette et al. (2016), these total $^{228}$Ra fluxes can be used to estimate fluxes of nutrients, iron and rare earth elements. $^{228}$Ra fluxes are also a
way to estimate the SGD fluxes by subtracting the contribution from rivers, diffusion and bioturbation, and dividing the remaining flux by the mean $^{228}$Ra concentration in groundwater. By this method SGD has been estimated at $0.03 - 0.48 \times 10^{13}$ m$^3$ yr$^{-1}$ in the Mediterranean Sea (Rodellas et al., 2015) and $2 - 4 \times 10^{13}$ m$^3$ yr$^{-1}$ in the Atlantic Ocean (Moore et al., 2008). These direct approaches suffer from strong potential biases associated with the relative sparsity of observations: raw assumptions have to be made in order to estimate regional averages (Rodellas et al., 2015) or to interpolate scattered observations in space
(Moore et al., 2008). Therefore, it is suitable only in regions with dense sampling, such as the Atlantic basin.





Inverse modeling techniques represent an alternative and powerful approach to estimate the fluxes, providing that the ocean circulation is known with sufficient accuracy. Inverse modeling is based on three elements: tracer (Ra) observations, a forward prognostic model which simulates the tracer distribution as a function of the fluxes to be assessed (here the spatial distribution and intensity of the shelf fluxes), and an algorithm optimizing the fluxes of the model in order to minimize the misfit between

the observations and the model results. The advantage of this method is that the ocean model is used as an interpolator, based on physical considerations, which is expected to be robust and consistent. Kwon et al. (2014) used such an inverse modeling approach to produce a global estimate of $^{228}$Ra fluxes of $9.1 - 10.1 \times 10^{23}$ atoms yr$^{-1}$ between 60°S and 70°N, corresponding to $9 - 15 \times 10^{13}$ m$^3$ yr$^{-1}$ of SGD. Their study is based on a data-constrained global ocean circulation model (DeVries and Primeau, 2011), considering 50 source regions on the continental shelf, and minimizing an ordinary least-squares cost function.

In this study, we estimate the radium fluxes from all continental shelves around the world and localize the most intense sources, using an inverse modeling technique with more data than previous studies (Kwon et al., 2014). The data set has been augmented with data from two recent GEOTRACES cruises and from the Southern Ocean, the North Pacific, the Mediterranean Sea and the Indonesian Seas. It also contains data from the Arctic, a basin absent from Kwon's study. The forward model is built on the Ocean General Circulation Model (OGCM) NEMO. Our main improvement is a careful analysis of sensitivity and errors,

which reveals that the result depends on the model mathematical parameters, such as the cost function and the number of regional sources that are considered. We have performed several inversions and analyzed the residuals and uncertainties, to determine the most appropriate mathematical parameters and evaluate the precision of the flux estimates. This paper is organized as follows. In section 2, we describe the different aspects of the inversion technique, e.g. the global dataset, the forward model based on the NEMO OGCM, the different cost functions, the choice of the source regions related to SGDs, and the inverse

method. Section 3 presents the main results, e.g. the global and regional estimates of $^{228}$Ra supply, and the sensitivity of these estimates to several parameters of our approach, such as the cost function or the number of coastal $^{228}$Ra sources. Section 4 compares our results with results obtained in previous studies, and discusses issues associated with such an approach, with an emphasis on SGD.

## 2   Methods

### 2.1   $^{228}$Ra dataset

Since the late 1960's (Moore, 1969; Kaufman et al., 1973) tens of oceanographic cruises have carried out measurements of $^{228}$Ra (e.g. articles listed in Table S1). The dataset used in this study includes, among others, observations from three international programs sampling trace elements. Data from the Indian Ocean cruise of GEOSECS, in 1977-1978, are included. From 1981 to 1989, the TTO (Transient Tracers in the Oceans) program produced a considerable number of $^{228}$Ra measurements in

the Atlantic, from 80°N to 60°S, at all depths, making of the Atlantic Ocean the best sampled ocean by far. Currently, new data from all oceans are being produced by GEOTRACES. 6059 data from all basins are used in total, of which 2789 are shallower than 10 m, 1107 are located between 10 and 200 m deep, 606 between 200 and 600 m deep, and 1557 deeper than 600 m. Our data set comprises 1359 more measurements than in Kwon et al. [2014]: Two GEOTRACES sections, GA03 (United States



– Cape Verde – Portugal) and GP16 (Ecuador – French Polynesia), and data from the Arctic, the Southern Ocean, the North Pacific, the Mediterranean Sea and the Indonesian Seas have been added in the present study (See Table S1). These supplementary observations extend the data coverage to regions north of 70°N and around the Kerguelen islands. In a near future, other sections from the GEOTRACES program will complete the global covering and may help in studying deeper sources.

For the purpose of our study, data have been averaged in each model grid cell (see section 2.2 for more details on the model configuration), leading to the 3076 cell averages shown on Fig. 1. The density of the measurement is noticeably uneven. The Atlantic Ocean, north of 20°S, and the Arctic, are the most densely covered basins and the only regions with a significant number of data at depth deeper than 10 m. Other regions are sparsely sampled, leaving wide areas with no or very few measurements, like the western Indian Ocean, the equatorial Pacific or the Pacific sector of the Southern Ocean.

Data are expressed in concentration or activity units, with the following conversion factor: $1 \, \mathrm{dpm \, m^{-3}} = 4.36 \times 10^{6} \, \mathrm{atoms \, m^{-3}}$. They range from $0.04$ to $724.5 \, \mathrm{dpm \, m^{-3}}$. The highest concentrations are found in the Bay of Bengal and the coastal seas of eastern Asia, the lowest values are located in the Southern Ocean. Concentrations are generally higher than $10 \, \mathrm{dpm \, m^{-3}}$ in the Indian Ocean, the Atlantic and the Pacific north of 30°N, lower in the rest of the Pacific. In the Atlantic, west of a line running from the Amazon delta to Newfoundland, most concentrations are higher than $30 \, \mathrm{dpm \, m^{-3}}$.

## 2.2 Forward tracer model

The second requirement of the inversion technique is a $^{228}$Ra transport model, allowing to link in situ observations to the boundary conditions or shelf sources of $^{228}$Ra. The transport equation of tracer $A_i$ (originating from the source region i) is given by:

$$\frac{\partial A_i}{\partial t} = -\boldsymbol{U}.\boldsymbol{\nabla} A_i + \boldsymbol{\nabla}.(K\boldsymbol{\nabla} A_i) - \lambda A_i + S_i \qquad (1)$$

$U$ and $K$ are the velocity field and the eddy diffusivity coefficient respectively. The two first terms on the right together constitute transport. They are derived from NEMO 3.6 model (Nucleus for European Modeling of the Ocean) using OPA (Madec, 2015) as a general circulation component, coupled with the sea-ice model LIM3 (Vancoppenolle et al., 2009), with an ORCA2 global configuration. The model has an horizontal resolution of $2° \times 2° \cos\phi$ (where $\phi$ is the latitude) enhanced to $0.5°$ near the equator. The mesh is tripolar in order to overcome singularities, the North Pole being replaced by two inland poles in the Northern Hemisphere. It has 31 vertical levels, ranging from the surface to 6000 m deep, the upper layer covering the first 10 m. The simulation is forced by a seasonal climatological dataset, based on NCEP/NCAR reanalysis and satellite data.

$\lambda$ is the radioactive decay constant, $0.12 \, \mathrm{yr^{-1}}$, given by the half-life of $^{228}$Ra which is $5.75$ yr. Decay is the sole sink. It is known and does not depend on environmental parameters, leaving the source term as the only unknown to be determined by

the inversion technique.

$S_i$ is the source term specific to the $\mathrm{i^{th}}$ region, representing riverine inputs, sedimentary diffusive fluxes and groundwater discharge fluxes of $^{228}$Ra from region i. In this study, sources are assumed to be only on the continental shelf, defined as the seabed shallower than 200 m. This depth range, spanning 16 model levels, is chosen because it is where most groundwater




discharge outflows. However, diffusion from sediments also occurs at higher depths (Hammond et al., 1990).

Equation (1) shows that the $A_i$ fields depend linearly on $S_i$. That means that any $^{228}$Ra distribution can be written as a linear combination of the $A_i$ fields. It is important to emphasize that the circulation is supposed to be "perfect", e.g. no correction of the $U$ field is looked for. Nevertheless, the simulated circulation obviously suffers from deficiencies, and that point has to

be kept in mind when interpreting the results. From now, what we refer to as "model concentration", $[^{228}Ra]^{mod}$ , is a linear combination of the tracer final concentrations $A_i$, the coefficients being the source intensities $x_i$:

$$[^{228}Ra]^{mod} = \sum_{i=1}^{n} A_i x_i \qquad (2)$$

A non-optimal model concentration was computed by assuming a uniform constant flux per unit of surface everywhere, with a global fit using the average concentration estimated from the observations: This defines the first guess estimate before the

inversion. The inversion undertaken in this study aims at optimizing the parameters $x_i$ in order to minimize the total difference between the observations and the model $^{228}$Ra distribution.

As a consequence of its coarse horizontal resolution, continental shelves are only poorly resolved by the ORCA2 grid. The emitting surface is underestimated and some regions with narrow continental shelves would be completely omitted. To overcome that deficiency, sub-model grid scale bathymetric variations are accounted for by comparing the model grid to a global

$2'$ resolution bathymetry ETOPO2 of the National Geophysical Data Center (NGDC). The algorithm is detailed in Aumont and Bopp (2006). According to this method, the total surface of continental shelf is $2.73 \times 10^{13}$ m$^2$, 73% higher than the $1.58 \times 10^{13}$ m$^2$ obtained with the coarser bathymetry.

The ocean – continent interface, including the Arctic and the Antarctic, is divided into 38 regions (Fig. 2). This first guess takes into account the sampling coverage (very low in the Antarctic for instance, and higher in the North Atlantic Basin or

Bay of Bengal) and differences in the tracer distributions $A_i$, which should be large enough to give independent information. Delimitation is done by trials and errors, using the posterior covariance matrix of the inversion (see section 2.3): The number of sources is minimized by merging regions associated with negligible fluxes with close highly correlated regions. Most islands are ignored, because the areas of their continental shelf and thus their expected contributions to the $^{228}$Ra balance are small. In the inversion process, islands can give rise to spurious fluxes to accommodate for other types of errors. The only islands

considered in this study are the Kerguelen and Crozet Islands, in the Southern Ocean, because many samples have been taken in their surroundings which make it possible to constrain their contributions. Because of the lack of measurements and the coarse model resolution, the Persian Gulf, the Red Sea, the Baltic Sea, the North Sea and the Hudson Bay are not taken into account. The flux per unit of surface is assumed to be constant on each of the 38 emitting regions. Model simulations last for the equivalent of 100 years in order to reach a quasi steady state. It is more than 17 times larger than the half-life of $^{228}$Ra,

so that the total amount of $^{228}$Ra does vary by less than 0.001%. As there are not enough data to study global inter-annual or seasonal variations, we do not take seasonal variations of $^{228}$Ra concentrations into account. We implicitly assume that radium concentration is constant over time, and work with average concentrations over the $100^{th}$ year of simulation.





## 2.3 Inverse method

The last requirement of the inversion technique is to define a cost function measuring the misfit between the data and the model. This cost is then minimized by a method already used to assess air-sea gas fluxes (Gloor et al., 2001; Mikaloff Fletcher et al., 2006; Jacobson et al., 2007) and oceanic heat fluxes (Resplandy et al., 2016). If $\eta$ represents the residuals, e.g. the discrepancy

between model results and observations:

$$\mathbf{A}\boldsymbol{x} = [\mathbf{^{228}Ra}]^{\boldsymbol{obs}} + \boldsymbol{\eta} \qquad (3)$$

Both sides are vectors, their size being the number of data. A is the matrix of footprints, representing the circulation model and composed of the $A_i$ at each data point. $x$ is the vector of unknowns, the flux of $^{228}$Ra per unit of surface of each region. As only shelf sources are modeled and as data coverage below $200 \, \mathrm{m}$ is sparse, only data shallower than $200 \, \mathrm{m}$ are considered.

The distance between model concentrations and data is summed up in a scalar, the cost function $C(\boldsymbol{x})$. We look for the optimal flux vector $\mathrm{x}_{opt}$ minimizing the cost function. Different choices for $C$ are possible, depending on the assumed probability distribution for the prior error on data and model footprints $\mathbf{A}$. Errors due to biased sampling are not considered here. All errors are supposed to be uncorrelated. In this study, three different cost functions have been tested and minimized. They all correspond to the sum of squares of a specific type of residuals. Their respective equations are listed in Table 1. The first one

is an ordinary least-squares cost function, $C_{ols}$. According to the Gauss-Markov theorem, its minimization produces the best linear unbiased estimator when prior errors have no correlation, zero expectation, and the same variance. It is the simplest least-squares method, chosen in the study by Kwon et al. (2014). This function gives the same weight to all observations. However, the hypothesis of the homogeneity of the variance is questionable: Far offshore, where concentrations are lower and less sensitive to small changes in coastal sources, observations and model errors can be expected to be lower. Neglecting

this fact means these data are not fully exploited, as their contribution to the cost function is relatively small. Two other cost functions with a higher weight for smaller values are then considered for comparison. They are assuming heteroscedastic data, with higher variances for higher concentrations. In the proportional least-squares cost function, $C_{prp}$, the error standard deviation is supposed to be proportional to the observed concentrations. The logarithmic least-squares cost function, $C_{log}$, works differently and assumes the logarithms of concentrations have the same error variance. It is less sensitive than $C_{prp}$

to model overestimations and more to underestimations. It is the only cost function which is not a quadratic function of the sources.

The residuals after inversion indicate what the inverse model cannot fit. In a "perfect" inversion, these residuals should be assimilated to noise, e.g. small and without structure, due for instance to coarse resolution. In most inversions, that is not the case, and the distribution of residuals emphasizes biases or errors either in the chosen hypotheses (such as a perfect

circulation) or in the setting of the inversion (number and choice of the regions). The posterior uncertainties on radium fluxes and correlations between regions are computed following the method described in Appendix A. A regional flux has a large uncertainty when it is constrained by few data or correlated to other regions (Gloor et al., 2001), and two regions are strongly correlated when the $^{228}$Ra emitted by each is transported to the same places and are then harder to differentiate. The computed posterior uncertainties are precise only if all the preceding assumptions on prior errors are correct. The coherence of error





assumptions with the results has to be checked (see 3.2). Along with the main inversion considering 38 regions, we performed 4 other inversions with a higher or lower number of regions in order to estimate the sensitivity to this parameter (see 3.3).

## 3   Results

### 3.1   $^{228}$Ra fluxes

The $^{228}$Ra fluxes from each of the 38 regions, deduced by minimizing each of the three cost functions, are shown on Fig. 3 with their confidence intervals, and compared with the prior estimates. The global fluxes for each method are also shown. As they are sums of local fluxes, their standard deviations are proportionally lower.

The global $^{228}$Ra flux within one standard deviation is $8.01 - 8.49 \times 10^{23}$ atoms yr$^{-1}$ according to the $C_{log}$ inversion. As we will explain in section 3.2., this estimate is the most accurate of the three. Fluxes are found to be comparatively high in the

North Atlantic (regions 5 to 16), in the western Pacific (22 to 27) and in the Indian Ocean (28 to 34), together accounting for 62.6% of the continental shelf and 82.8% of the global flux of $^{228}$Ra. Highest fluxes are located on the east coast of North America (10 and 13), in the China Seas (23 and 25) and in the eastern Indian Ocean (29 to 32), where the inversion process produces the largest increase compared to prior estimates. Conversely, inversion significantly reduces the prior estimates in the Arctic Ocean (35 to 37), in the Bering Sea (21) and in the eastern Pacific Ocean (17 to 20). Fluxes are also quite low in the

Southern Ocean (1 and 38) and in the South Atlantic (2 to 4). The newly estimated Arctic and Antarctic sources are in the range of 0.43 to 0.50 and 0.31 to $0.37 \times 10^{23}$ atoms yr$^{-1}$ respectively, accounting for 5% to 6.2% and 3.6% to 4.6% of the total sources. In total, roughly two thirds of the $^{228}$Ra flows into the Indian and Pacific basins, contrasting with the 60% of the global river discharge flowing into the Atlantic and Arctic basins (Milliman, 2001).

Although having the same order of magnitude, uncertainties are generally lower than fluxes. They are highest in the western

Pacific and Indian Oceans (regions 22 to 34), because of data sparsity. It is lower in better sampled oceans: The Arctic (35 to 37) and the Atlantic (2 to 16) Oceans, except for region 13 (Cape Hatteras to Newfoundland). The eastern Pacific (17 to 21) also has low uncertainties in absolute values, probably due to the low concentrations and prior errors there.

The two other inversions produce roughly similar results, although fluxes are generally lower when derived from $C_{prp}$ and generally have higher uncertainties when derived from $C_{ols}$. The global $^{228}$Ra flux is estimated to $7.16 - 8.14 \times 10^{23}$ atoms yr$^{-1}$

with $C_{ols}$ and $4.96 - 5.28 \times 10^{23}$ atoms yr$^{-1}$ with $C_{prp}$. The three inversions agree on which basins and continents have the largest and smallest sources. Yet, local disagreements occur. Regions 5 (Amazon delta), 21 (Alaska and Bering Sea) and 33 (Arabian Sea) have higher fluxes with $C_{ols}$ than with $C_{log}$, with non-overlapping confidence intervals, whereas the contrary is true for regions 12 (Mediterranean), 26 (Indonesian Seas) and 34 (East Africa). $C_{prp}$ fluxes are generally lower, and in eight regions their confidence intervals overlap with none of the other inversions. Discrepancies happen because the fitting of the

model to each observation implies different and possibly opposite effects on the source intensities. Each inversion uses different weights, which translates into different flux corrections. Fluxes from each of the three inversions are most dissimilar for regions where observations impose most dissimilar constraints, where the model fails to reconstruct the pattern of the data and has to choose between fitting some data or others in priority. In such cases, all the results should be considered carefully. When





the confidence intervals between the different inversion techniques fail to overlap, it is likely that one or several estimates are incorrect. As algorithms are built by assuming a prior error statistics, it is likely that some rely on wrong assumptions.

## 3.2 Model concentrations and residuals

The residuals, i.e. the differences between model concentrations and observations, determine how well the model reproduces
the observations and quantify the improvements in the tracer distribution provided by the inversion. They are also a basic tool to identify biases in the model and to assess the quality of the assumptions.

Radium fluxes obtained by the inverse method largely improve the model match to observations compared to the prior radium flux (Fig. 4 and Fig. 5). The improvement is quantified by the increase in the model–data correlations (Table 2) and the decrease in the root mean square of the residuals (Table 3), a proxy of the cost function. The correlation coefficient is increased from
0.383 to 0.813 on a linear scale and from 0.809 to 0.902 on a logarithmic scale. The correlation is higher on a logarithmic scale because it is less sensitive to the few very high residuals associated with the highest concentrations (see Fig. 7). On average, the inversion is able to reduce the ordinary residuals ($C_{ols}$) by a factor 2, logarithmic residuals ($C_{log}$) by 1.4 and proportional residuals ($C_{prp}$) by 3.5.

In spite of being smaller, the order of magnitude of the residuals remains comparable to the data (Fig. 5). On the one hand,
in all oceans, positive and negative residuals are observed with no clear patterns at the scale of a few grid cells. Because of the rather low model resolution ($2°$, which is not sufficient to reproduce medium and small scale processes) and of issues associated with temporal (seasonal or higher frequency) variability of the data, this kind of "noise" is expected. It is consistent with the assumption of independent errors used when computing the error variances on fluxes. On the other hand, in several regions, residuals can display coherent large scale patterns, which cannot be attributed to noise. These areas may suffer from
systematic overestimations, like in the Gulf Stream region, the western Pacific between $20°$N and $40°$N, and off Eastern Siberia, or underestimations, such as in the center of the North Atlantic Gyre. These residuals point out to possible flaws in the model circulation. For instance, $^{228}$Ra is quite homogeneously distributed in the western North Atlantic according to data, but in the model, the gradients are stronger, and no combination of sources manages to reduce these gradients. This is probably caused by a bias in the Atlantic circulation, with a too low exchange rate between inshore and offshore waters. In a $2°$ resolution model,
mesoscale eddies are not represented and cannot transport $^{228}$Ra south and east of the Gulf Stream or north of the North Brazil Current. Inversions minimize the misfit by increasing the fluxes from regions 5 (North Brazil), 8 (Caribbean) and 10 (southern East Coast of the US), making the model concentration too high close to the coast while still too low in the gyre. Such large scale biases are not consistent with the assumption of no prior error correlation, which may lead to underestimation of flux uncertainties around these basins.
Having assumed specific prior error statistics when choosing the cost functions, we need to check that there is no a posteriori contradiction. Figure 6 displays the residuals and model concentrations as a function of the observations. If the residuals depend on the observed concentrations, it means some observations are more precise than others, contain more information, and should be given a higher weight in order to obtain the best linear unbiased estimate. Figure 7 shows the probability density functions (PDF) of residuals after all three inversions and compares them with a Gaussian curve representing the expected distribution





given the root mean square of residuals. Each PDF should look like a Gaussian curve for the computed posterior uncertainties to be relevant descriptors of errors.

Figure 7a emphasizes that the ordinary residuals do not follow a Gaussian distribution. On the contrary most residuals are very close to zero and only a small number of them is much higher than the standard deviation. Figure 6a shows that these high residuals occur at high concentrations only, and that error variance is not homogeneously distributed. Then high and low concentrations should not be given the same weights, as in $C_{ols}$, but the highest concentrations should be given the lowest weights, as in $C_{log}$ and $C_{prp}$. Flux estimates based on $C_{ols}$ are biased because the cost function puts more emphasis on high concentrations, and this method then tries to fit more specifically the misfits at high concentrations. $C_{ols}$ also produces very large error bars because the error variance is assumed to be constant and its computed value, influenced by a few very large residuals, is larger than the actual error variances of the vast majority of data. Figure 6c and Fig. 7c show that the proportional residuals are not normally distributed either. They are not even symmetrical, as they cannot be smaller than -1 but they do not have an upper limit. This asymmetry produces a bias in the flux estimate. The algorithm based on $C_{prp}$ is more sensitive to positive residuals because underestimations are never associated to proportional residuals lower than -1 whereas overestimations can produce residuals higher than 1. As a consequence, this method tends to reduce the fluxes. The hypothesis of constant variance is more realistic although the highest residuals occur at low concentrations. Finally, the distribution of the logarithmic residuals displayed on 6b) and 7b) is much closer to a Gaussian curve and much less dependent on concentrations, which makes the logarithmic cost function more relevant for this study.

## 3.3 Sensitivity to the number of regions

The choice of the regions (and their number) has been made rather subjectively, although several criteria have been used (spatial distribution of the observations, independence of the $A_i$ fields). The global $^{228}$Ra should ideally not depend upon the number of regions. Therefore, alternative region geometries have been tested for comparison. The $^{228}$Ra fluxes are shown on Table 4 and the root mean square of their residuals is presented on Table 5. Case 1 inversion uses 52 regions: It was the original distribution of regions before some of them were merged to define the 38 standard regions of this study. It includes more regions in undersampled areas such as the Arctic, the South Atlantic, the western Indian or the equatorial Pacific. Case 5 has just one emitting region for each of the following ocean basins: Southern, South Atlantic, North Atlantic, South Pacific, North Pacific, Indian and Arctic. Case 3 and Case 4 are intermediate cases with source regions built by merging regions from Case 2. The root mean square of residuals is a proxy of the cost function. On the one hand, this parameter should be as low as possible. Increasing the number of regions always decreases it because the number of degrees of freedom increases, which tends to improve the fit to the observations. In this inversion, the largest decrease is found between 7 and 12 source regions. Further increases in the number of regions have smaller impacts. On the other hand, too many source regions may produce spurious results. Some regional fluxes, with too few observations nearby to constrain them, would be computed using observations farther away, already used by other fluxes. Because of the lower sensitivity of the concentrations at these farther-off locations, this process can create extreme fluxes, positive or even negative. The presence of physically impossible negative values, set to zero by the constraint of positivity, necessarily means such poor constraints exist. When 52 fluxes are computed, 5 to 7 of





them, according to the cost function, are so poorly constrained that their fluxes have been set to zero to prevent them from being negative. This number is reduced to 1 with $C_{ols}$ and zero with $C_{log}$ and $C_{prp}$ when there are 38 regions, and completely disappears with 19 or fewer regions. Regions with fluxes within the error estimate are also very poorly constrained by the observations and the circulation model: their number is also reduced from 26 out of 52 to 11 out of 38 with $C_{ols}$, from 13 to 3

with $C_{prp}$ and from 9 to 3 with $C_{log}$. All these fluxes have a low impact on the cost function, but make the global $^{228}$Ra flux less precise. This analysis shows that Case 1 (with 52 regions) is not constrained enough and that Case 3 to 5 display too large residuals. Therefore Case 2 (38 regions) is considered to be the best choice.

The $C_{ols}$-based global $^{228}$Ra flux is varying in a non-monotonic way, with a difference of 23% between the highest and the lowest. With 38 regions, it is lower than fluxes computed with both a higher and a lower number of regions. This high

sensitivity may be related to the high uncertainty associated with this inversion, certainly linked to the relatively poor data coverage. All the confidence intervals within one standard deviation from Case 1 to Case 4 overlap. The $C_{prp}$-based global fluxes are always lower than the other fluxes, and they decrease as the number of regions decreases. This is consistent with our previous hypothesis on $C_{prp}$. This cost function tends to fit the lowest data in priority. Larger regions suffer more from this bias because they are constrained by more data, likely to be more dispersed. The confidence intervals within one standard deviation

based on $C_{prp}$ fail to overlap. Only the logarithmic least-squares method produces very similar fluxes whatever the number of regions, with all confidence intervals overlapping. The global flux based on $C_{log}$ again seems to be the most reliable.

### 3.4 Submarine groundwater discharge estimates

The shelf fluxes after inversion combine groundwater discharge, riverine particles, diffusion from sediments and bioturbation. Here we deduce the contribution from groundwater discharge by using existing estimates of the other sources of radium.

Rivers are poor in dissolved $^{228}$Ra and transport $^{228}$Ra mainly with the sediments they carry (Moore and Shaw, 2008). A fraction of them is desorbed in the mixing zone, because of the salinity increase. According to various studies (Key et al., 1985; Moore et al., 1995; Krest et al., 1999), the amount of $^{228}$Ra desorbed per gram of sediment lies in the range of 2.9 to $8.7 \times 10^6$ atoms. In this study, we follow Milliman (2001) who proposed a global river sediment flux of $1.8 \times 10^{16}$ g.yr$^{-1}$, divided into fluxes from six basins: South Atlantic, North Atlantic, South Pacific, North Pacific, Indian Ocean and Arctic. This

leads to a global $^{228}$Ra river flux estimate of $0.53 - 1.60 \times 10^{23}$ atoms yr$^{-1}$, significant but not dominant.

$^{228}$Ra is released from the sediments by diffusion, bioturbation and advection, the latter being associated with the SGD. Like Moore et al. (2008), we assume that the $^{228}$Ra fluxes by diffusion and bioturbation from relict sands, composing 70% of the total continental shelf area, is weak, typically of the order of $10^9$ atoms m$^{-2}$ yr$^{-1}$ (Colbert, 2004; Hancock et al., 2006). Fluxes from continental shelf muds, which correspond to the remaining 30%, have been estimated by several studies using different

methods, such as inventories (Moore et al., 1995), benthic chambers (Hancock et al., 2000), sediment profiles (Hancock et al., 2000) or modeling (Hancock et al., 2006). But some of them were done at a time when SGD were not considered to be an important source of radium and may have included them in their estimates. These estimates should then be considered as an upper limit. Furthermore, their locations are often very close to the coast. Some recent studies addressing diffusive fluxes separately in order to estimate local groundwater discharge (Crotwell and Moore, 2003; Kim et al., 2005) used the simplified equation of





diffusion from Krest et al. (1999) and pointed out to significantly lower values, but may not be representative of all continental shelves. So far, it is not possible to be very precise and a wide range has to be considered: From the low recent values, close to $5 \times 10^9$ atoms.m$^{-2}$.yr$^{-1}$, to the higher values typically ranging from 25 to $75 \times 10^9$ atoms m$^{-2}$ yr$^{-1}$ (Moore et al., 2008). The full range is then $5 - 75 \times 10^9$ atoms.m$^{-2}$.yr$^{-1}$. As the continental shelf area in our model is $2.73 \times 10^{13}$ m$^2$, this means a total flux due to diffusion and bioturbation ranging from 0.43 to $6.51 \times 10^{23}$ atoms yr$^{-1}$. Then, using the logarithmic cost function, the SGD $^{228}$Ra flux estimate varies between 0.62 and $6.82 \times 10^{23}$ atoms yr$^{-1}$: these two fluxes are within the same range.

Figure 8 shows the $^{228}$Ra fluxes for seven basins corresponding to the Southern Ocean and the six basins used by Milliman (2001) to define sediment inputs by rivers. The largest $^{228}$Ra fluxes are found in the North Atlantic, the North Pacific and the Indian Ocean, in roughly equal proportions, whereas those from the South Atlantic, the South Pacific and the Arctic Ocean are far smaller. Fluxes from the three latter regions are low enough in fact to be within the confidence intervals of riverine and diffusive sediment $^{228}$Ra fluxes, so that the SGD confidence intervals include negative values, which are physically impossible. SGD $^{228}$Ra end-member concentration varies considerably from one aquifer to another, ranging from 0.04 to $125 \times 10^6$ atoms m$^{-3}$ (Moore et al., 2008). Measurements so far have shown aquifer concentrations in the Atlantic Ocean higher than the average by typically 30% (Kwon et al., 2014). Following Kwon et al. (2014), we assume that the aquifer concentrations are log-normally distributed with an average of 0.98 to $1.15 \times 10^3$ dpm m$^3$. Taking a geometric mean rather than an arithmetic mean is implicitly assuming that aquifers characterized by high $^{228}$Ra concentrations are emitting less water. It also suffers from wells being concentrated in developed countries. Using the results from the inversion technique based on the logarithmic cost function, the total SGD flux is estimated to $1.3 - 14.7 \times 10^{13}$ m$^3$ yr$^{-1}$, to be compared with the global river flow of $3.5 \times 10^{13}$ m$^3$ yr$^{-1}$ (Milliman, 2001).

## 4 Discussion

### 4.1 Comparisons with previous studies

Our inversion estimates are in good agreement with previous regional studies of $^{228}$Ra based on inventories. The $^{228}$Ra inventory of the Mediterranean Sea computed by Rodellas et al. (2015) led to a total flux due to rivers, sediments and groundwater discharge of $1.86 - 2.48 \times 10^{22}$ atoms yr$^{-1}$. This is compatible with our estimate based on $C_{log}$, $1.96 - 2.28 \times 10^{22}$ atoms yr$^{-1}$. Other cost functions produce an underestimation in the model by a factor 2.5 (ols) or 3.8 (prp). In the Yellow Sea, Kim et al. (2005) estimated a flux of $3.3 \times 10^{15}$ dpm yr$^{-1}$, or $1.4 \times 10^{22}$ atoms yr$^{-1}$. Expressed per unit of surface, it corresponds to $3.6 \times 10^{10}$ atoms m$^{-2}$ yr$^{-1}$. This is slightly lower than our flux for the larger region 23 (Sea of Japan, Yellow Sea, East China Sea) of $4.2 - 5.6 \times 10^{10}$ atoms m$^{-2}$ yr$^{-1}$ based on $C_{log}$, both these fluxes being larger than the global average. At larger scale, Moore et al. (2008) estimated the total $^{228}$Ra flux over the Atlantic between 50°S and 80°N to be $2.8 - 4.2 \times 10^{23}$ atoms yr$^{-1}$. Restricting the inversion to the Atlantic with $C_{log}$ yields $2.64 - 2.92 \times 10^{23}$ atoms yr$^{-1}$, in the lower range but compatible. At least two reasons can explain the difference. Moore et al. (2008) have included data down to 1000 m, which allowed them to estimate $^{228}$Ra release from sediments down to that depth. The authors have estimated the sources between 200 m and 1000 m





deep to $0.13 - 0.37 \times 10^{22}$ atoms yr$^{-1}$. Furthermore, in order to compute the total $^{228}$Ra content of the Atlantic, they have performed a linear interpolation of the data, potentially leading to errors, especially in areas where measurements are sparse. At the global scale, Kwon et al. (2014) have used a method which is quite similar to ours and they have computed a total global $^{228}$Ra flux of $9.1 - 10.1 \times 10^{23}$ atoms yr$^{-1}$, of which between 4.2 and $7 \times 10^{23}$ atoms yr$^{-1}$ are released by SGD. This corre-

sponds to a global SGD flux of $9 - 15 \times 10^{13}$ m$^3$ yr$^{-1}$. The flux of $^{228}$Ra estimated by the present study is thus significantly lower than the estimates by Kwon et al. (2014), although our results include the Arctic and Antarctic sources ($0.43 - 0.50$ and $0.28 - 0.35 \times 10^{23}$ atoms yr$^{-1}$ respectively). Kwon et al. (2014) minimized an ordinary least-squares cost function with 50 regions. We have shown here that both a high number of source regions and the use of the ordinary least-squares cost function concur to produce a higher estimate. However, this is at the expense of a higher uncertainty and of producing unrealistic

negative fluxes in some source regions. Additional differences in Kwon's study may explain their higher estimates such as a different ocean circulation model, a coarser vertical resolution, and a bathymetry re-gridded onto the model domain. Finally, they added dust deposition and removed data higher than $140$ dpm m$^{-3}$, but these two factors should rather tend to reduce their fluxes.

As recently proposed by Charette et al. (2016), shelf $^{228}$Ra fluxes can be used as gauges of shelf fluxes of trace elements and

isotopes, including nutrients, iron, and rare earth elements. $^{228}$Ra is particularly relevant because it is chemically conservative and integrates information over annual to decadal timescales. At first approximation, the flux of TEIs is deduced from the $^{228}$Ra flux and the ratio of the nearshore gradients. Limited for now due to the lack of nearshore measurements, this method could be more common in the future. As they are based on realistic assumptions on error statistics and have low uncertainties, our radium fluxes are able to improve the current estimates of all elements originating from the continental shelf.

On the contrary, our uncertainties on groundwater discharge are large, even when compared to previous estimates. These larger uncertainties stem from the poor knowledge of the non SGD sources of radium that we subtract from the total flux. As diffusion and bioturbation are expressed in flux per area, the mean SGD fluxes and their uncertainties depend to a large extent on the radium emitting area we consider. Based on a more realistic refined bathymetry than Kwon et al. (2014) ($2.73 \times 10^{13}$ m$^2$ compared to $1.5 \times 10^{13}$ m$^2$), our study also has a larger sedimentary flux, with an upper range close to the total $^{228}$Ra flux.

Thus, the lower range of SGD fluxes, $1.3 \times 10^{13}$ m$^3$ yr$^{-1}$, is very low, while the upper range, $14.7 \times 10^{13}$ m$^3$ yr$^{-1}$, is similar to other studies. The use of this model provides an upper estimate but cannot precisely compute the global SGD flux, and no inverse model can if the surface diffusive flux and the area emitting radium are not clarified. Our expectation is that the lower range of sedimentary flux is more likely, because it comes from studies where SGD is also considered and because it does never produce negative regional fluxes. By contrast, the higher range is very similar to the total bottom flux, as expected when

no distinction is made between diffusion and SGD.

Comparisons with local direct estimates of submarine groundwater discharge based on seepage meters and piezometers are also possible but less conclusive because of the high spatial variability of SGD. Our average SGD fluxes are between 0.5 and $5.5$ m yr$^{-1}$. Values from seepage meters and piezometers in the upper 200m reported by Taniguchi et al. (2002) and Knee and Paytan (2011) range from $0.03$ m yr$^{-1}$ in the Tokyo Bay (Taniguchi et al., 2002) to $1790$ m yr$^{-1}$ near Mauritius (Burnett

et al., 2006). Most measurements have been performed in the upper 10 meters and range from 1 to 50 m.yr$^{-1}$. Thus, this range





is very wide and many local studies span several orders of magnitude. At a global scale, Taniguchi et al. (2009) produced an estimate of the global SGD flux based on seepage measurements of $6.1 - 12.8 \times 10^{13}$ m$^3$ yr$^{-1}$, in the upper range of our global estimate. However, it suffers from most measurements being concentrated in developed countries.

## 4.2 Model biases

Our model of $^{228}$Ra is based on a circulation model and assumptions on the cycle of this isotope. Both are potential sources of errors.

Dust has not been included in the model. The model then replaces them with other sources, potentially leading to an overestimation. At global scale, it represents $1.7 \times 10^{21}$ atoms yr$^{-1}$ (Kwon et al., 2014), less than 0.2% of the total flux and less than most individual source regions, which cannot create large biases. Nevertheless, the largest dust deposition is associated to

Saharan dust transported to the North Atlantic, in the Canary Upwelling Region (Mahowald et al., 2005). In this region, dust may have some impact on the $^{228}$Ra distribution, since it brings this isotope directly to the open ocean. Its absence in our study might explain the overestimation by our model near the coast of region 6 and underestimation offshore. Yet, the exclusion of dust deposition in our analysis cannot explain the largest bias in the North Atlantic: The overestimation in the Gulf Stream coupled with an underestimation just south east (see 3.2), because the area where the model cannot transport radium is located

west of the maximum of dust input and displays higher concentration.

Scavenging is a neglected potential sink in this study. As the residence time related to scavenging is approximately 500 yr (Moore and Dymond, 1991), it accounts for between 1% and 2% of all sinks. Thus, its inclusion in our computations would increase the source intensity necessary to maintain global balance. Contrary to radioactivity, scavenging is highly heterogeneous, most intense where primary productivity and particle concentrations are highest. Fluxes from the high latitudes of the

North Atlantic are thus potentially more underestimated than fluxes in other regions, since they are areas of intense biological activity during spring blooms. As the actual total lifetime of $^{228}$Ra is overestimated when scavenging is not taken into account, the gradient between coast and open ocean could be too low. However, as the horizontal mixing time scale of the ocean is a few decades, the relative overestimation of open ocean concentrations is less than 10%.

The contribution of rivers to radium fluxes is considered when estimating the SGD, but only at a basin-wide scale. As most

riverine $^{228}$Ra travels attached to particles and is desorbed in the mixing zone, we have based our computation on the sediment loads of Milliman (2001), which are basin-wide. Although NEMO 3.6 takes rivers into account for their impact on salinity (Madec, 2015), for now no information at the model grid scale is available on their sediment loads. If this information existed, we would have been able to estimate the contribution of rivers, and consequently sediments and SGD, in the $^{228}$Ra fluxes for each of the 38 regions. Some local SGD fluxes close to large rivers (Amazon, Congo, Yellow River, etc.) could then appear to

be significantly lower than their shares of the total $^{228}$Ra fluxes suggest.

The other part of the model is the circulation model. The climatological circulation of NEMO 3.6 was not optimized in this study. However, the residuals after inversion show that some regions are associated with spatially structured residuals. There are good reasons to incriminate the ocean circulation. Because of the low resolution of the model (2°), isopycnal diffusion has been used to parameterize sub grid processes and mixing (Redi, 1982) with a constant eddy diffusivity of 2000 m$^2$ s$^{-1}$. Nev-



ertheless, in very energetic regions, such as the Western Boundary Currents (WBCs, e.g. the Gulf Stream and the Kuroshio), higher eddy diffusivity might be needed to enhance the exchanges with the open ocean, possibly improving the fit with the observations. Furthermore, it is well known that the mean currents are also dependent on this low resolution (impacting for instance the position and intensity of the WBCs). Although it cannot solve all the flaws of the circulation, improvements could thus be brought by an increase of the resolution towards eddy resolving models. However, as 100 years of simulation are needed at a global scale for each source region, this would be computationally expensive.

Other sources of errors are the four statistical assumptions on the errors: errors are assumed to have zero expectation, no correlation, a normal distribution and variances depending on the concentrations in a way specific to each cost function. Systematic biases on data or model are not corrected by least-squares algorithms. They increase or decrease values without leaving clues. However, model conserves mass: the quantity of radium present in the ocean from each model tracer is precisely known and if concentration is too high at some place, it will be too low elsewhere. As for the measurements, their uncertainty is generally around 10% or lower, and cruises take them independently from each other, making the assumption of zero expectation on the observational errors reasonable. The second assumption is the absence of prior correlation. If prior uncertainties of neighboring data are correlated, it means that the errors are likely to be of the same sign whatever the solution, and that multiplying measurements in this area does not multiply information proportionally. Where measurements are dense, with residuals far from the expected white noise, for instance in the North Atlantic, there may be correlation and uncertainties may be underestimated. The last assumptions are about the structure of variance. We have shown that logarithmic residuals were almost normally distributed and independent from concentrations (see 3.2.), justifying the choice of $C_{log}$ as a cost function. $C_{ols}$ leads to higher uncertainties, especially for small sources, and $C_{prp}$ to systematic underestimation, but both are useful for comparison, in order to identify regions where physical assumptions are inaccurate (see 3.1.).

### 4.3    What new data would be most useful ?

Observations are not evenly distributed. Some coastal regions cannot be constrained properly because $^{228}$Ra data are lacking. Improving the coverage would increase the quality of the inversion in two ways: It would reduce the uncertainties and make it possible to divide wide regions into smaller regions as long as they have distinct footprints. For instance, the Philippines, Papua, or the Gulf of Guinea, whose footprints are very different from the Indonesian Seas and the southwestern coast of Africa could be considered as independent regions. More samples in the Indian Ocean, the South Atlantic (south of $30°$S), the Southern Ocean and the western equatorial Pacific are priorities. All these regions will be sampled by upcoming GEOTRACES cruises shown on Fig. 8. At the same time, deep samples will be taken outside of the Atlantic, enabling a more comprehensive global inversion with extra source regions at greater depths. This would improve our knowledge of the contribution of deeper sediments.

Most direct submarine groundwater discharge measurements have been performed in developed countries, with a focus on the North Atlantic and the Mediterranean Sea (Taniguchi et al., 2002). At the same time, measurements are completely lacking over large regions. More SGD studies in areas where they are potentially the highest, namely the Bay of Bengal, the Indonesian Seas and the China Seas, would produce more representative estimates of the $^{228}$Ra content of SGD around the world and direct





estimates of local SGD magnitude to be compared with regional inversion results. They have begun more recently (Knee and Paytan, 2011) and are still sparse.

Information contained in $^{228}$Ra might be completed with $^{226}$Ra concentrations, measured during the same campaigns and for this reason available with a similar coverage. Associated with the same source as $^{228}$Ra, but with a much longer half-life,

1602 yr, $^{226}$Ra would constrain deeper sources and would help in assessing the quality of the thermohaline circulation and deep ventilation of the circulation model. However, inverting $^{226}$Ra data would require a precise modeling of scavenging, which is not negligible at these longer time scales, as well as a much longer integration duration in order to get steady state distribution.

## 5 Conclusions

Based on inverse modeling, we have computed a global $^{228}$Ra flux from continental shelves of $8.01 - 8.49 \times 10^{23} \ \mathrm{atoms \ yr^{-1}}$, with the largest sources in the western Pacific, the western North Atlantic and the Indian Oceans and the smallest sources in the eastern Pacific. The Arctic and Antarctic sources have been estimated for the first time, accounting for $0.43 - 0.50$ and $0.28 - 0.35 \times 10^{23} \ \mathrm{atoms \ yr^{-1}}$ respectively. These precise estimates are obtained by minimizing the squares of the differences of logarithmic concentrations between model and data, a cost function we think is more realistic than the other functions we

have tested because it is based on more realistic assumptions on error statistics. Given the number of available measurements, we were able to constrain 38 regional fluxes. The shelf fluxes produced using these optimal parameters are lower than previous estimates. In a near future, they will enable to quantify continental shelf fluxes of trace elements and isotopes to the oceans at any place where nearshore gradients are measured (Charette et al., 2016). The estimated global SGD flux is far less precise, ranging between 1.3 and $14.7 \times 10^{13} \ \mathrm{m^3 \ yr^{-1}}$, because of the very large uncertainty on the two other sources of $^{228}$Ra, i.e.

riverine particles and most of all diffusion from bottom sediments, also located on the continental shelf. Only the upper range is compatible with previous estimates. After inversion, we were able to reproduce the basin scale patterns of $^{228}$Ra distribution with nevertheless systematic biases in several regions, especially in the Arctic, and west of the sub-tropical gyres. Shortcomings in the circulation model are the most probable explanation of these biases (too weak exchanges between continental shelves and open ocean). Therefore, besides estimating the sources and sinks of tracers, interesting information about the ocean model

can be obtained by Ra like tracer inversions. Extensive regions are lacking observations, mostly in the Southern Hemisphere (Pacific, Indian, and mostly Southern Ocean, as well as western equatorial Pacific Ocean), and better coverage in basins where SGDs are known to be influential and to produce large horizontal gradients is also needed (such as in South Asia). But the main impediment to achieve precise estimates of global Ra SGD fluxes comes from the very poor knowledge of diffusive sedimentary fluxes: without a proper way to separate diffusion and SGD, inversions can compute the total bottom flux but are

not able to precisely evaluate these two components.





## 6 Code and Data availability

The source code of NEMO is available on the NEMO website. Studies providing data are listed in Table S1 in the supplementary materials. The inversion code and data used in this study can be obtained directly by contacting the authors.

## Appendix A:  Error Statistics

This section describes the way regional $^{228}$Ra flux estimates and their error bars are computed. The inverse problem is the following one : using p measurements of oceanic $^{228}$Ra concentrations, their n sources must be traced back, by means of a circulation model. The model produces n $^{228}$Ra concentration fields, one from each of the source regions. It is run through steady state, the i$^{\text{th}}$ field reaching concentration $A_i$. The model concentration is a linear combination of the latter:

$$[^{228}Ra]^{mod} = \sum_{i=1}^{n} A_i x_i$$

Inversion requires fitting the model concentration to the observations. As the expression of model concentration is linear in terms of the unknowns, the problem can be written:

$$\mathbf{A}\boldsymbol{x} = \boldsymbol{b} + \boldsymbol{\eta}$$

where $\boldsymbol{b} \in \mathcal{M}_{p,1}$ is the vector of the p measured radium concentrations, $\boldsymbol{x} \in \mathcal{M}_{n,1}$ is the unknown and the vector of the n source intensities, $\mathbf{A} \in \mathcal{M}_{p,n}$ is the footprint matrix, composed of the $A_i$ at each data point, and $\boldsymbol{\eta} \in \mathcal{M}_{p,1}$ is the vector of

residuals, accounting for the misfits, which are inevitable in our over-constrained problem (p>n).

We want to minimize the distance between data and model concentrations, summed up in a cost function. The ordinary least-squares cost function we first used is given by:

$$C_{ols}(\boldsymbol{x}) = (\mathbf{A}\boldsymbol{x} - \boldsymbol{b})^{\top}(\mathbf{A}\boldsymbol{x} - \boldsymbol{b})$$

The cost function has to be minimal at the optimal $x$, called $x_{opt}$, implying, in the absence of inequality constraints:

$$\boldsymbol{\nabla} C_{ols}(\boldsymbol{x_{opt}}) = 2\mathbf{A}^{\top}(\mathbf{A}\boldsymbol{x} - \boldsymbol{b}) = \mathbf{0}$$

As the n $^{228}$Ra concentration fields are independent from each other, $A^{\top}A$ is invertible. Then:

$$\boldsymbol{x_{opt}} = \mathbf{A_{inv}}\boldsymbol{b} \tag{A1}$$

$$\mathbf{A_{inv}} = (\mathbf{A}^{\top}\mathbf{A})^{-1}\mathbf{A}^{\top}\boldsymbol{b}$$

$\mathbf{A_{inv}}$ is the pseudo-inverse of $\mathbf{A}$, transforming the consequences into their most probable causes and justifying the word "inversion". Equation (A1) corresponds to (2.95) in Wunsch (2006). In practice, fluxes are constrained to be positive. This is





managed by just reducing the number of source regions if necessary, without changing the principle of the inversion.

Matrix $\mathbf{H}$ si defined by :

$$\mathbf{H} = \mathbf{A}(\mathbf{A}^\top\mathbf{A})^{-1}\mathbf{A}^\top = \mathbf{A}\mathbf{A}_{\mathbf{inv}}$$

As $\mathbf{H^2} = \mathbf{H}^\top = \mathbf{H}$, $\mathbf{H}$ is an orthogonal projector on a subspace of dimension n. As $\mathbf{A}x_{opt} = \mathbf{H}b$, the model concentration is the projection of the observations on this subspace whereas the vector of residuals is the projection on the complementary subspace.

Simple algebraic transformations yield:

$$\mathbf{HA} = \mathbf{A}(\mathbf{A}^\top\mathbf{A})^{-1}\mathbf{A}^\top\mathbf{A} = \mathbf{A}$$
$$\mathbf{A}_{\mathbf{inv}}\mathbf{A} = (\mathbf{A}^\top\mathbf{A})^{-1}\mathbf{A}^\top\mathbf{A} = \mathbf{I_n}$$
$$\mathbf{A}_{\mathbf{inv}}\mathbf{A}_{\mathbf{inv}}^\top = (\mathbf{A}^\top\mathbf{A})^{-1}\mathbf{A}^\top\mathbf{A}(\mathbf{A}^\top\mathbf{A})^{-1} = (\mathbf{A}^\top\mathbf{A})^{-1} \tag{A2}$$

Uncertainties on $x$ resulting from uncertainties on data ($b$) and model ($\mathbf{A}$) must now be evaluated. By principle of the ordinary least-squares cost function, the error covariance of $(\mathbf{A}x - b)$ is considered to be a constant diagonal matrix:

$$\langle(\mathbf{A}x - b)(\mathbf{A}x - b)^\top\rangle = \sigma^2\mathbf{I_p}$$

The posterior covariance matrix of $x$ is:

$$\begin{aligned}
\mathbf{V_{xx}} &= \langle(x - x_{opt})(x - x_{opt})^\top\rangle \\
&= \langle(\mathbf{A}_{\mathbf{inv}}\mathbf{A}x - \mathbf{A}_{\mathbf{inv}}b)(\mathbf{A}_{\mathbf{inv}}\mathbf{A}x - \mathbf{A}_{\mathbf{inv}}b)^\top\rangle \\
&= \mathbf{A}_{\mathbf{inv}}\langle(\mathbf{A}x - b)(\mathbf{A}x - b)^\top\rangle\mathbf{A}_{\mathbf{inv}}^\top \\
&= \sigma^2\mathbf{A}_{\mathbf{inv}}\mathbf{A}_{\mathbf{inv}}^\top \tag{A3}
\end{aligned}$$

Equation (A3) corresponds to (2.102) in Wunsch (2006).

$\sigma^2$ is related to the root mean square of residuals $v_\eta$ the following way:

$$\begin{aligned}
v_\eta &= \langle(\mathbf{A}x_{opt} - b)^\top(\mathbf{A}x_{opt} - b)\rangle \\
&= \langle(\mathbf{H}b - b)^\top(\mathbf{H}b - b)\rangle \\
&= \langle((\mathbf{I_p} - \mathbf{H})(\mathbf{A}x - b) + (\mathbf{HA} - \mathbf{A})x)^\top((\mathbf{I_p} - \mathbf{H})(\mathbf{A}x - b) + (\mathbf{HA} - \mathbf{A})x)\rangle \\
&= \langle(\mathbf{A}x - b)^\top(\mathbf{I_p} - \mathbf{H})(\mathbf{A}x - b)\rangle \\
&= \sigma^2\frac{p - n}{p} \tag{A4}
\end{aligned}$$

Then, using Eqs. (A2), (A3) and (A4), we can compute $\mathbf{V_{xx}}$ :

$$\mathbf{V_{xx}} = \frac{p}{p - n}v_\eta(\mathbf{A}^\top\mathbf{A})^{-1} \tag{A5}$$





The diagonal terms of $\mathbf{V_{xx}}$ are the squared uncertainties on $\boldsymbol{x}$ whereas the other terms are covariances, indicating source regions with similar footprints.

The result can be extended to cases where the cost function is weighted, such as the proportional least-squares inversion, by just normalizing matrix $\mathbf{A}$ and vector $\boldsymbol{b}$. Extension to logarithmic inversion is done by linearizing $\log(\mathbf{A}\boldsymbol{x}) - \log(\boldsymbol{b})$ to $\mathbf{F}\boldsymbol{x} - \boldsymbol{\beta}$

5   with $\boldsymbol{\beta} = \log(\boldsymbol{b}) + \mathbf{F}\boldsymbol{x_{opt}} - \log(\mathbf{A}\boldsymbol{x_{opt}})$ , assuming a small error.

*Competing interests.*  The authors declare that they have no conflict of interest.

*Acknowledgements.*  The authors thank all the scientists who produced the data used in this article. We thank Matt Charette, Eun Young Kwon, Virginie Sanial and Pieter van-Beek for helping us putting the data together. We also thank Olivier Marchal for a discussion about algorithms. This work is part of the first author's PhD, supported by the "Laboratoire d'Excellence" LabexMER (ANR-10-LABX-19) and
10  co-funded by a grant from the French government under the program "Investissements d'Avenir", and by a grant from the Regional Council of Brittany.





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





**Table 1.** Cost functions

| Cost function | Formula | Residuals |
|---|---|---|
| Ordinary least-squares (ols) | $C_{ols}(x) = \sum\limits_{j=1}^{p} \left([^{228}Ra]_i^{mod} - [^{228}Ra]_i^{obs}\right)^2$ | $\eta_{ols}(x) = [^{228}Ra]^{mod} - [^{228}Ra]^{obs}$ |
| Logarithmic least-squares (log) | $C_{log}(x) = \sum\limits_{j=1}^{p} \left(\log[^{228}Ra]_i^{mod} - \log[^{228}Ra]_i^{obs}\right)^2$ | $\eta_{log}(x) = \log[^{228}Ra]^{mod} - \log[^{228}Ra]^{obs}$ |
| Proportional least-squares (prp) | $C_{prp}(x) = \sum\limits_{j=1}^{p} \left(\frac{[^{228}Ra]_i^{mod} - [^{228}Ra]_i^{obs}}{[^{228}Ra]_i^{obs}}\right)^2$ | $\eta_{prp}(x) = \frac{[^{228}Ra]^{mod} - [^{228}Ra]^{obs}}{[^{228}Ra]^{obs}}$ |

**Table 2.** Correlation between data and model concentration fields

| Model | Linear correlation | Logarithmic correlation |
|---|---|---|
| Prior | 0.383 | 0.809 |
| ols | 0.813 | 0.883 |
| log | 0.797 | 0.902 |
| prp | 0.754 | 0.877 |

**Table 3.** Root mean square of residuals before and after inversion

| Model | Ordinary residuals rms (dpm.m$^{-3}$) | Logarithmic residuals rms (no unit) | Proportional residuals rms (no unit) |
|---|---|---|---|
| Prior | 70.7 | 0.892 | 1.91 |
| ols | 36.6 | 0.703 | 1.10 |
| log | 37.9 | 0.636 | 1.02 |
| prp | 47.4 | 0.940 | 0.558 |

**Table 4.** Model global $^{228}$Ra fluxes ($10^{23}$ atoms yr$^{-1}$) with different numbers of source regions

| Cost function | Case 1: 52 regions | **Case 2: 38 regions** | Case 3: 19 regions | Case 4: 12 regions | Case 5: 7 regions |
|---|---|---|---|---|---|
| ols | $7.81 - 9.93$ | **$7.16 - 8.14$** | $7.51 - 8.49$ | $7.36 - 8.16$ | $8.98 - 9.78$ |
| log | $8.09 - 8.61$ | **$8.01 - 8.49$** | $7.90 - 8.34$ | $7.85 - 8.27$ | $7.93 - 8.37$ |
| prp | $5.15 - 5.47$ | **$4.96 - 5.28$** | $4.70 - 4.98$ | $4.25 - 4.49$ | $3.75 - 4.01$ |





**Table 5.** Root mean square of residuals after inversions with different numbers of source regions

| Cost function | Case 1: 52 regions | **Case 2: 38 regions** | Case 3: 19 regions | Case 4: 12 regions | Case 5: 7 regions |
|---|---|---|---|---|---|
| ols | 35.8 | **36.6** | 37.6 | 38.2 | 42.9 |
| log | 0.623 | **0.636** | 0.656 | 0.671 | 0.781 |
| prp | 0.545 | **0.558** | 0.581 | 0.597 | 0.665 |





**Figure 1.** Observed concentrations of $^{228}$Ra in the global ocean, averaged when available on the ORCA2 cells used for inversion. Data plotted on subfigures b), c) and d) are also vertically averaged in order to take all layers into account.



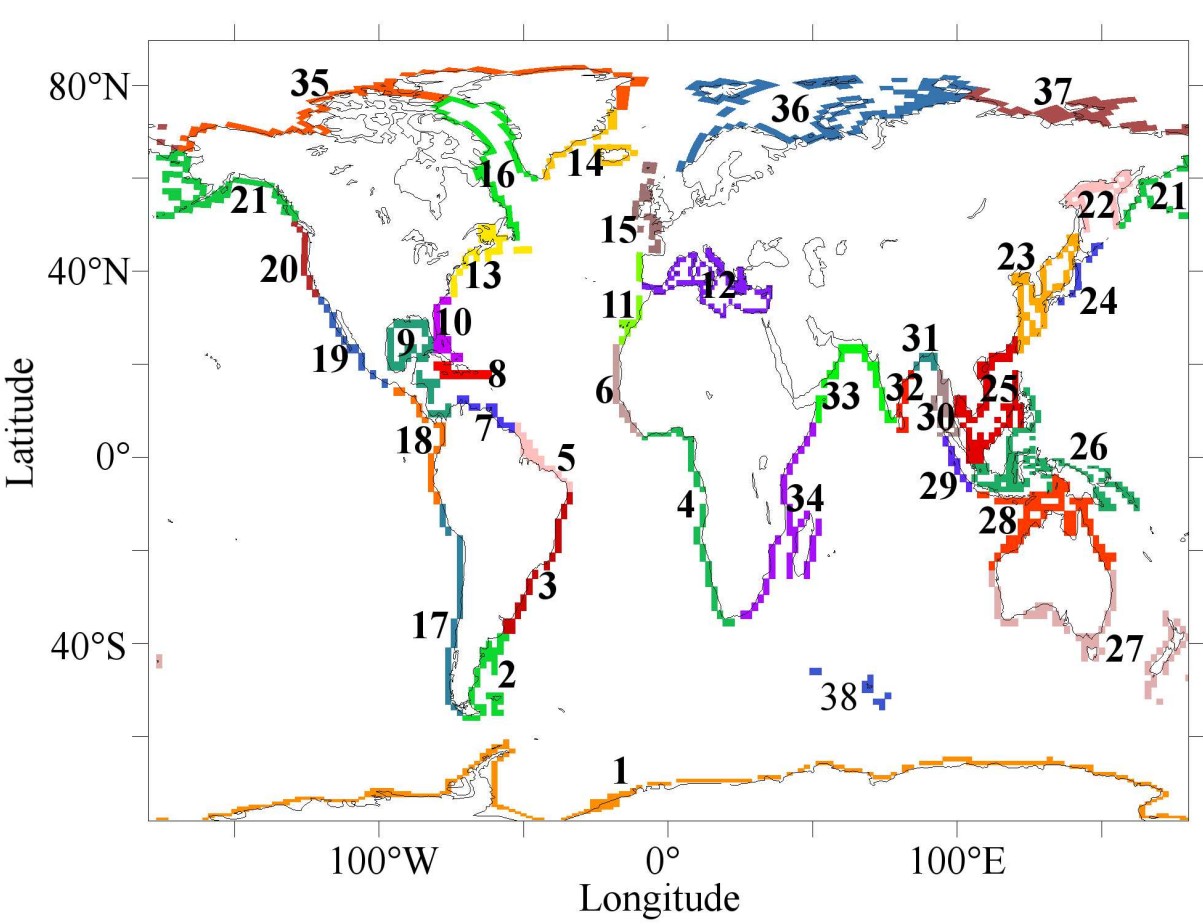

**Figure 2.** Model $^{228}$Ra source regions





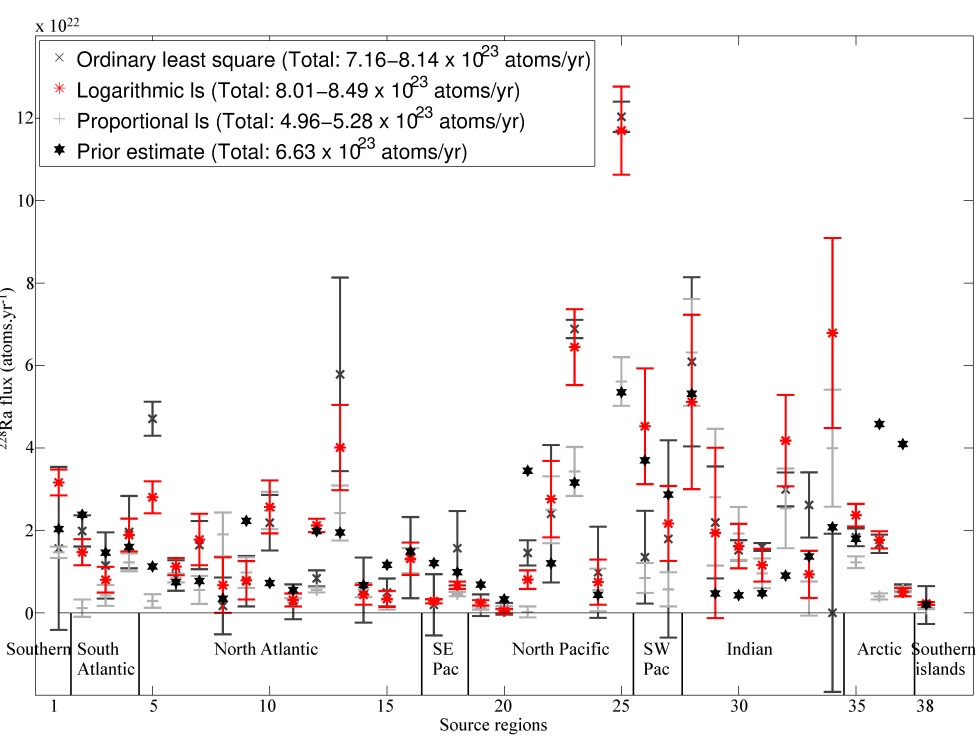

**Figure 3.** $^{228}$Ra annual flux from each source within one standard deviation, after minimization of three cost functions. Prior estimates, proportional to shelf surfaces, are shown for comparison.





**Figure 4.** Model surface $^{228}$Ra concentration after minimization of three cost functions. Prior estimate is shown for comparison.





**Figure 5.** Surface $^{228}$Ra logarithmic residuals after minimization of three cost functions. Prior estimate is shown for comparison.





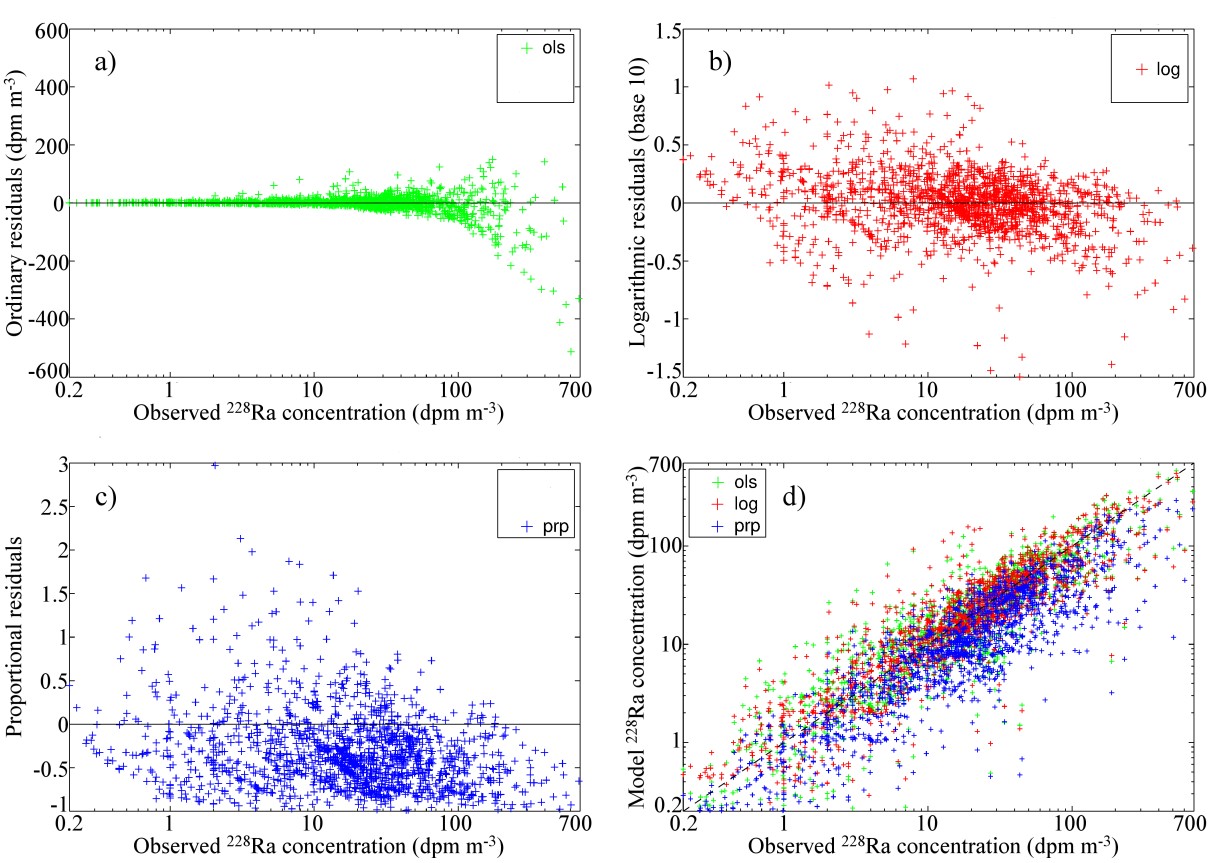

**Figure 6.** (a) Residuals, (b) logarithmic residuals, (c) proportional residuals and (d) model $^{228}$Ra concentration as a function of observed $^{228}$Ra concentration. Lines of zero residuals are drawn in black.





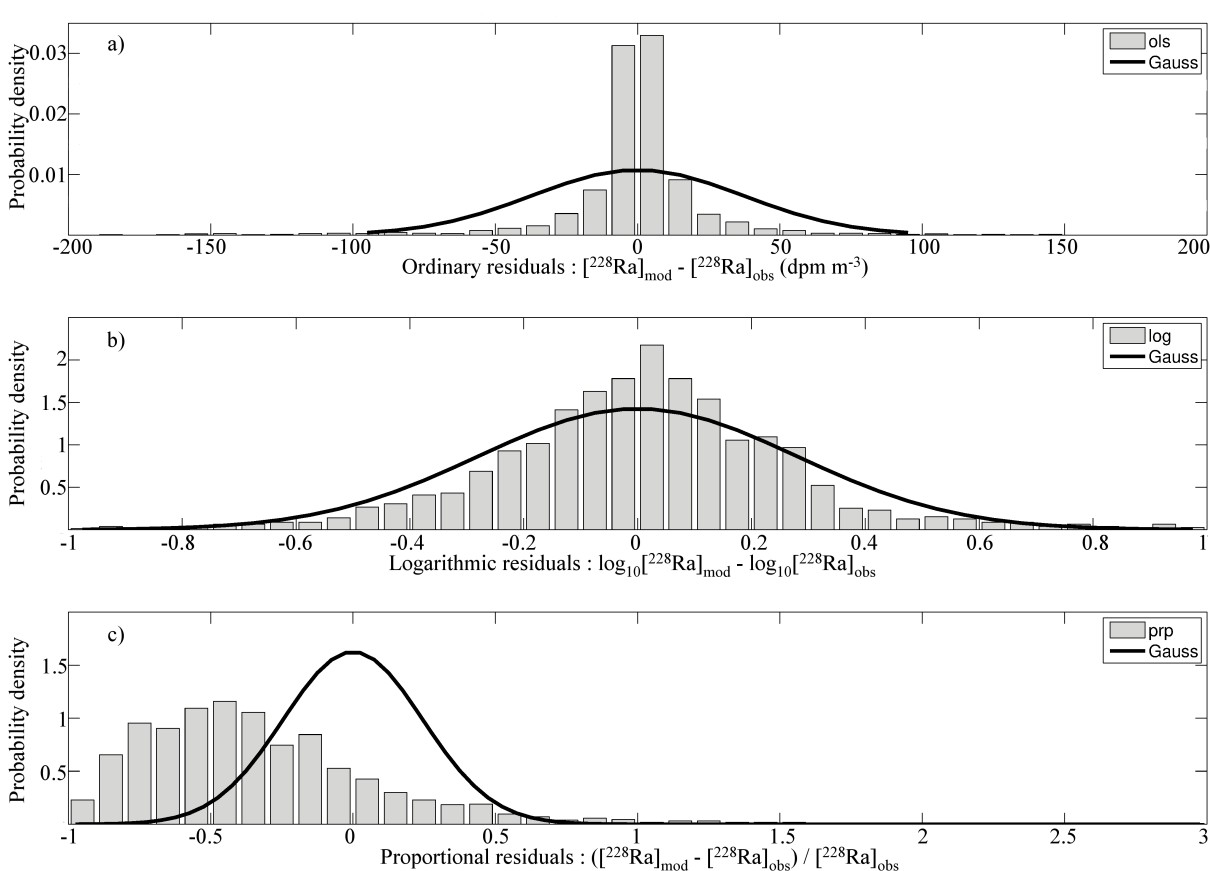

**Figure 7.** Probability Density Functions (PDF) of $^{228}$Ra (a) ordinary residuals, (b) logarithmic residuals and (c) proportional residuals after inversion, compared with a Gaussian PDF (black full line) based on their standard deviations.





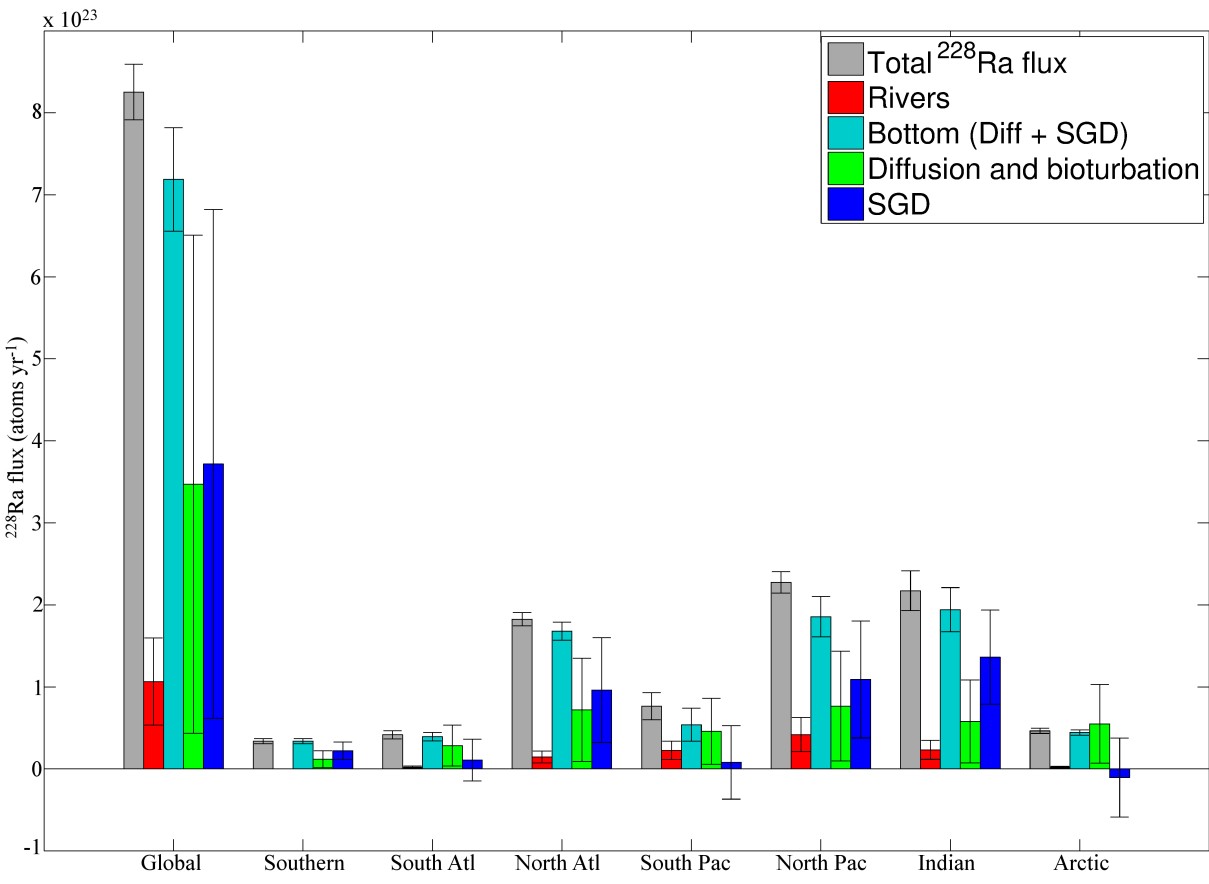

**Figure 8.** $^{228}$Ra sources by basin, divided into riverine input, diffusion from sediment and SGD. Only logarithmic least-squares results are shown. Fluxes are divided the following way: Regions 1 and 38: Southern; $2 - 4$: South Atl; $5 - 16$: North Atl; $17 - 18$ and $26 - 27$: South Pac; $19 - 25$: North Pac; $28 - 34$: Indian; $35 - 37$: Arctic





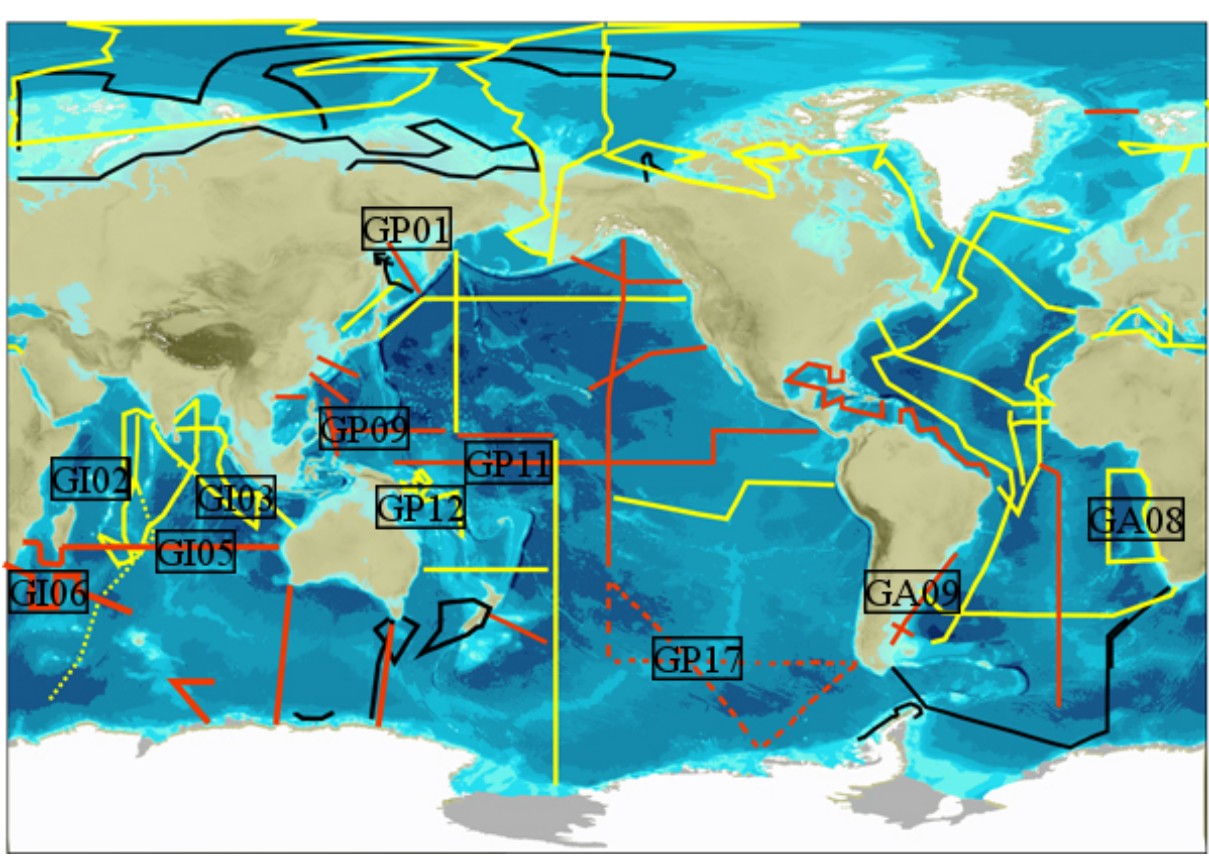

**Figure 9.** map of GEOTRACES cruises (from http://www.geotraces.org/cruises/cruise-summary). Planned sections are in red, completed sections in yellow, International Polar Year sections in black. Names in rectangles correspond to cruises potentially bringing the largest extra information on $^{228}$Ra fluxes because they are performed in places lacking measurements.