# Peer review of "Improving the inverse modeling of a trace isotope: how precisely can radium-228 fluxes toward the ocean and Submarine Groundwater Discharge be estimated?"

_Biogeosciences, 2017_

## Short Comment (SC1) · 22 Feb 2017

This is an outstanding paper making excellent progress in a timely and important field. The authors used an excellent approach to reduce uncertainty in Ra-228 global budgets. I feel the authors should be congratulated and the paper should be published after minor review. I have only a few minor suggestions for improvement: 1) Page 1, Line 2, abstract: "lower". Add a short note on how much lower. 2) Page 2, Line 33: "raw assumptions". I suggest the authors spell out the major raw assumptions here, or just omit this early criticism. 3) Page 7, Line 17: Why the Indian and Pacific Basins are so high? I looked for a comment on that later in the discussion but could not find. I encourage the authors to add a paragraph (probably in the discussion) offering some

thoughts explaining the spatial distribution. 4) What depth of the upper ocean was used to integrate the radium observations? How does it compare to previous studies? What sort of extrapolation was made in terms of depth integration for locations with no data in deeper waters? 5) Page 10, Line 21: "A fraction" can be replaced by "Nearly all". 6) Page 10, Line 30: A number of diffusion studies are briefly cited. Considering the emphasis on diffusion, I encourage the authors to add more information about those studies. Maybe a summary table with the source of diffusion data and how it was estimated. 7) Page 11, Line 20 and elsewhere: The comparison between SGD and river flow is appropriate and should be kept since it puts results in perspective. However, radium-derived SGD is likely to be saline water, while rivers are a source of fresh water. I encourage the authors to add a note qualifying those differences using the literature. 8) Similar to the previous comment, the comparison to seepage meters on the last paragraph of page 12 may need to be qualified. Many seepage meter deployments are made in very shallow nearshore waters and may capture fresher SGD, while Ra-228 covers a much larger scale. Consider using radium studies to build this comparison. Overall, this is an excellent study that should be published with no delay. My comments are mostly suggestions for improvement rather than conditions for acceptance.

---

## Referee Comment (RC2) · Anonymous Referee #2 · 20 Mar 2017

This manuscript describes the use of an inverse modeling technique to investigate 228Ra fluxes to ocean basins. The inverse modeling technique itself is not novel, nor is its application to studying 228Ra fluxes, however the efforts taken by these authors to examine each of the assumptions of the model is valuable and results in improved estimates of the 228Ra fluxes to each basin. This work also includes flux estimates for the Arctic and Southern Oceans, which have not been included in previous studies. Because of these two valuable contributions I recommend that this paper be published after revisions to address the specific comments below.

Specific comments: 1. I do not think the title accurately represents the main focus of the paper, which is an effort to improve the application of the inverse modeling technique

[Figure]

to 228Ra, as opposed to an interpretation of an updated 228Ra flux estimate or SGD estimates. Either the title should be changed to better reflect the fact that the main contributions of this work are improvements to the model, or more discussion should be added on the interpretation of the results of the model and how they change our understanding of 228Ra and/or SGD. If the latter is chosen, the authors should be sure to highlight how their contributions are unique from those of Kwon et al (2014), aside from simply reporting the newly calculated Arctic and Southern Ocean 228Ra fluxes. 2. Page 2: There is a short discussion about the sources of Ra, and it is mentioned that dust inputs are small compared to the other sources used in the model. The dissolved riverine source should also be introduced here instead of later in the paper, because this is where the sources/sinks are first introduced. You state later on that the dissolved component of the riverine flux is negligible, but it would be better to approximate the relative contribution as you have done for the dust source in line 20. 3. Page 7 lines 17 – 18: Why is the flux of 228Ra (in units of atoms y-1) compared to river discharge (I'm assuming this is units of m3 y-1)? Why not compare the flux of SGD to river discharge, so that it is a volume-to-volume comparison? I'm not sure if this is intended to make the point that the largest flux of 228Ra does not correlate with the greatest river discharge, but the authors argue that rivers do not carry much dissolved 228Ra, so in that case they shouldn't correlate anyway. 4. Page 8 line 11: Should this be a reference to figure 6 instead of figure 7? If you do indeed mean to call figure 7, then the figures should be re-ordered so that they are in the order they are referenced in the paper (i.e. don't reference figure 7 before figure 6 is referenced for the first time). 5. Page 14 line 28: should call figure 9 instead of figure 8 6. Conclusion lines 14 – 15: be specific about the other cost functions that were tested, instead of saying "…the other cost functions…". That way if a reader skips to the conclusion, they will still understand your specific results. The same sentence ends by saying "…more realistic assumptions on error statistics"; again, be specific about what the more realistic assumptions were. A large part of the paper is dedicated to testing assumptions, so the conclusion should point out the results of those tests. 7. Conclusion lines 24 – 25: The sentence that

begins with "Therefore, besides estimating the sources…" should not be included in the conclusion, as it is not one of the main points emphasized in the paper. It is more appropriate to move this line to the end of the "model biases" section or another part of the manuscript. 8. The conclusion should have a stronger ending and remind the reader why it is important to better resolve the 228Ra model. Line 17 is a good reminder of why determining the shelf fluxes is important, but the conclusion should also highlight why it is important to separate the SGD and diffusion sources. This could also be discussed in more detail earlier in the paper, but should at least be mentioned again in the conclusion. 9. Page 16 line 2: include the actual NEMO website 10. Full references for the data sources used in the model are not included in the reference list; is this because they are only listed in the supplementary material? I suggest adding a supplementary reference list for these data sources. Also, in the Arctic references, the year for the Rutgers van der Loeff et al. 2013 reference should be 2012, not 2013. 11. On Figure 9, the cruise across the Fram Strait has been completed, so this should be in yellow instead of red.

Technical corrections: In general, I recommend proofreading the manuscript again for grammatical errors and typos to allow for ease of reading. I have included a few examples here: 1. Sentences should not start with numbers or abbreviations (e.g. page 3 line 31). In many cases "228Ra" is the first word in a sentence; when this is the case it should be spelled out as "Radium-228". 2. Abbreviations should be defined at their first usage (for example, Ra should be defined in the last line of page 1 instead of 2, and NEMO should be defined on page 3 instead of 4). 3. Page 1 line 21: insert the word "are" in between "or" and "pollutants" so the sentence reads "…or are pollutants" 4. Page 2 lines 11-12: use the pronoun "its" instead of "their" 5. Page 3 line 30: delete the word "of" so the sentence reads "…making the Atlantic Ocean…" 6. Page 14 line 10: insert the word "the" before "model" so the sentence reads "However, the model conserves mass…" 7. Page 17 line 2: "si" should be "is"

---

## Author Comment (AC1) · 2 May 2017

Dear referee #2 Thank you for the thorough reading of our manuscript and your constructive comments. As requested, the manuscript will be proofread again for typos and grammatical errors. Here are the answers to your comments:

1) The title should reflect three aspects: a) We have shown that parameters such as the number of source regions and the cost function matter a lot in inverse modeling, b) By using appropriate parameters, we produce a more precise estimate than previous studies, c) However, this is not immediately usable for SGD studies as this source still have to be separated from diffusion. The current title might not emphasize the first aspect enough but finding a better one is difficult. The title could be changed to

"Improving the inverse modeling of a trace isotope: how well does it constrain radium-228 fluxes toward the ocean and Submarine Groundwater Discharge?".

2) Riverine dissolved radium-228 is less than 1% of the total radium-228 flux. This figure, similar to that of dust, will be added in the introduction (Page 2 line 30). More explanations will be given at page 10, using the annual river discharge and the average concentration in dissolved radium.

3) Page 7 line 17. It would indeed make more sense to compare river discharge to the other large fluxes of water and nutrients to the ocean, Submarine Groundwater Discharge (SGD), rather than to radium-228 fluxes. The problem is that SGD suffers from high uncertainty, as highlighted in the article. Average basin-wide SGD fluxes can be compared, but this is implicitly assuming that diffusion per unit of surface is nearly the same in both basins, which is far from certain. This issue will be moved to the end of 3.4, where we will compare the distribution of river discharge with both radium fluxes and SGD, and explain why the latter is imprecise, although more interesting. The ratios are very close anyway.

4) Page 8 line 11: You are right: the figure called here is Fig. 6. It will be changed and this change will solve the numbering problem at the same time.

5) Page 14 line 28: Indeed, the figure called here is Fig. 9.

6) Conclusion lines 14-15: The difference between the three cost functions and between the underlying assumptions on error statistics will be described explicitly in the conclusion, so that a reader skipping to the conclusion can understand what it is about and why it is important.

7) Conclusion lines 24-25: This sentence will be removed from the conclusion, as it is not a main focus of the paper. This possibility of improving a circulation model using the residuals of inversions is already mentioned in 3.2 and 4.2 and there is not much to add.

8) Evaluating the SGD is the goal of several studies and radium-228 has often been used as a tracer for this purpose. One important conclusion of this article is that using radium-228 in an inventory or in inverse model is very imprecise because SGD and diffusion are poorly separated. This will be stated more explicitly in 3.4 and in the conclusion.

9) NEMO website will be included (http://www.nemo-ocean.eu/).

10) The references of the data sources will be included in the supplement.

11) The Fram Strait section will be corrected.

---

## Author Comment (AC2) · 10 May 2017

Dear Isaac Santos

Thank you for the thorough reading of our manuscript and your constructive comments. Here are the answers to your comments:

1) The radium flux of our study is around 20% lower than the inventory of radium 228 in the Atlantic by Moore et al. [2008] and the global estimate of Kwon et al. [2014], which is significant. This will be stated in the abstract and the conclusion. The possible reasons are discussed in 4.1.

2) Page 2 line 33: This sentence is ambiguous and will be rephrased. No assumption

is required to perform box averages or interpolations. The "raw assumptions" are the very fact that box averages or linear interpolations can produce a precise and unbiased estimate of the total flux, which is not true when data are sparse.

3) This is a very interesting question. I think there are two points here: a) At page 7 line 17, a comparison is made between radium fluxes, a proxy of SGD, and rivers. My point is that these two sources of water and nutrients are distributed in a very different way. But, given the size of the basins, rivers may be more biased than radium/SGD. The Indo-Pacific basin accounts for 52% of continental shelves and 63% of radium fluxes, compared to 29% and 27% for the Atlantic. In order to address a comment by referee #2, this comparison will be moved to 3.4, transformed into a comparison between rivers and SGD, and the previous remark will be added among other improvements. b) The distribution of fluxes is more complex than "high in the Indian and Pacific, low in the Atlantic and Arctic". Differences within each basin are larger. Except for a few cases where the explanation is obvious, for instance the very low fluxes in the eastern Pacific, where the shelves are very thin, it is difficult to identify the reasons why some regions emit more radium than others. This is probably related to the local geology: high fluxes may be due to more permeable sediments, to more wave, tide or geothermal energy inducing larger SGD, but also to a higher content in radium-228 and its parent radionuclide, thorium-232. We are not specialists of this field and the literature we know so far does not provide the global scale pattern of these parameters. For instance, it does not explain what the main differences between regions like the Bay of Bengal or the Gulf of Guinea are. Previous radium and SGD studies do not explain the reason for the source distribution either. A few explanations on the potential causes will be added in the discussion, but relating differences in radium-228 fluxes to a specific geological parameter is not possible.

4) There may be a misunderstanding. No integration (averaging) on the water column is performed. So no extrapolation has to be made on concentrations between observations. This is the strength of inverse modeling. We just minimize the differences between observed concentrations and model concentrations at all observation points shallower than 200m, without assuming anything about radium concentration elsewhere or regional averages. This will be made clearer in the introduction, where some sentences of the current manuscript are ambiguous, especially on page 3 from lines 2 to 6.

5) Page 10 line 21: "A fraction" will be replaced by "nearly all", and more detail will be given about the small amount of riverine dissolved radium.

6) Page 10 line 30: More information will be added on the methods used to compute diffusion from sediments. Their results will be summed up in a table. It will show that some methods produce high values, close to the average shelf fluxes.

7) Page 11 line 20: Indeed, SGD is mainly recirculated saline water, as already explained in the introduction. According to Taniguchi [2007], there is only 2600 km3/yr of fresh SGD, between 2% and 20% of the total SGD. The comparison with rivers is made because they both contain nutrients and trace elements. The end of 3.4 will be changed to explain all this more clearly, using the literature.

8) Indeed, seepage meters are not representative of total SGD, as they are concentrated close to the coast, where both fresh and saline SGD are expected to be higher than the continental shelf average. This is why the fluxes of most local studies are one order of magnitude higher than ours. We will explain this more clearly. In the current version of the manuscript, the only mentioned weakness of seepage meters is spatial variability, which is incomplete. However, seepage meters are interesting because they are independent from radium and are not affected by uncertainties on radium diffusion or radium concentration in groundwater. Not mentioning them would be strange. Radium studies, at a larger scale, are already the object of the previous paragraphs.

---

## Author Response (AR1)

Dear Jack Middelburg,

Thank you for considering our manuscript for publication after minor revisions. We also thank Isaac Santos and Referee #2 for their time and effort reviewing this manuscript. Please find our answers to the comments, detailed changes and new manuscript, where changes are shown in red.

**Referee #1 (Isaac Santos)**

Comment 1: Page 1, Line 2, abstract: "lower". Add a short note on how much lower.

Answer 1: The radium flux of our study is around 20% lower than the inventory of radium 228 in the Atlantic by Moore et al. [2008] and the global estimate of Kwon et al. [2014], which is significant. This is now stated in the introduction (page 1 line 7) and in the conclusion (page 16 line 8)

Comment 2: Page 2, Line 33: "raw assumptions". I suggest the authors spell out the major raw assumptions here, or just omit this early criticism.

Answer 2: This sentence is ambiguous. No assumption Interactive is required to perform box averages or interpolations. The "raw assumptions" are the very fact that box averages or linear interpolations can produce a precise and unbiased estimate of the total flux, which is not true when data are sparse. It is has rephrased as: "These direct approaches suffer from strong potential biases when the data are sparse, because the total amount of $^{228}$Ra in the ocean is then roughly estimated, using regional averages (Rodellas et al., 2015) of observations or linear interpolations (Moore et al., 2008). Therefore, they are suitable only in regions with dense sampling, such as the Atlantic basin."

Comment 3: Page 7, Line 17: Why the Indian and Pacific Basins are so high? I looked for a C1 BGD Interactive comment Printer-friendly version Discussion paper comment on that later in the discussion but could not find. I encourage the authors to add a paragraph (probably in the discussion) offering some thoughts explaining the spatial distribution.

Answer 3: This is a very interesting question. I think there are two points here: a) At page 7 line 17, a comparison is made between radium fluxes, a proxy of SGD, and rivers. My point is that these two sources of water and nutrients are distributed in a very different way. But, given the size of the basins, rivers may be more biased than radium/SGD. The Indo-Pacific basin accounts for 52% of continental shelves and 63% of radium fluxes, compared to 29% and 27% for the Atlantic. In order to address a comment by referee #2, this comparison will be moved to 3.4, transformed into a comparison between rivers and SGD, and the previous remark will be added among other improvements. b) The distribution of fluxes is more complex than "high in the Indian and Pacific, low in the Atlantic and Arctic". Differences within each basin are larger. Except for a few cases where the explanation is obvious, for instance the very low fluxes in the

eastern Pacific, where the shelves are very thin, it is difficult to identify the reasons why some regions emit more radium than others. This is probably related to the local geology: high fluxes may be due to more permeable sediments, to more wave, tide or geothermal energy inducing larger SGD, but also to a higher content in radium-228 and its parent radionuclide, thorium-232. We are not specialists of this field and the literature we know so far does not provide the global scale pattern of these parameters. For instance, it does not explain what the main differences between regions like the Bay of Bengal or the Gulf of Guinea are. Previous radium and SGD studies do not explain the reason for the source distribution either. Relating differences in radium-228 fluxes to a specific geological parameter is not possible. However, a few explanations has been added in the discussion (page 13 lines 18-25): "The spatial distribution of the fluxes in this study is consistent with Kwon et al. (2014), with two thirds of radium-228 flowing to the Indian and Pacific Basins and also quite high fluxes in the western North Atlantic. It turns out that these basins have the 20 highest riverine sediment loads (Milliman, 2001). However, we have shown that rivers could account for 7% to 20% of $^{228}$Ra fluxes only, which does not exclude an indirect impact on the other sources through the geology of the continental shelf. A difference among basins in the quantity of radium diffusion, which is poorly known, is also possible. If it is not the case, then the SGD are significantly higher in East Asia than in other regions. SGD can be driven by as many factors as storms, waves, tides and thermal gradients, and depend on the permeability and structure of coastal and shelf sediments (Moore, 2010a). 25 Comparative data on these factors in several basins will be necessary to explain the origin of the observed differences."

Comment 4: What depth of the upper ocean was used to integrate the radium observations? How does it compare to previous studies? What sort of extrapolation was made in terms of depth integration for locations with no data in deeper waters?

Answer 4: There may be a misunderstanding. No integration (averaging) on the water column is performed. So no extrapolation has to be made on concentrations between observations. This is the strength of inverse modeling. We just minimize the differences between observed concentrations and model concentrations at all observation points shallower than 200m, without assuming anything about radium concentration elsewhere or regional averages. This will be made clearer in the introduction, where some sentences of the current manuscript are ambiguous, especially on page 3 from lines 2 to 6, which now read as : "The advantage of this method is that no arbitrary averaging or interpolation is required: observed concentrations are simply compared to model concentrations at the same points. It is expected to be robust and consistent, since the model is based on physical considerations."

Comment 5: Page 10, Line 21: "A fraction" can be replaced by "Nearly all".

Answer 5: Done

Comment 6: Page 10, Line 30: A number of diffusion studies are briefly cited. Considering the emphasis on diffusion, I encourage the authors to add more information about those studies. Maybe a summary table with the source of diffusion data and how it was estimated.

Answer 6: A sixth table was added, citing studies on diffusion from fine-grained sediments (those who contribute the most), their locations, their methods and the results. We have put in bold those which might have accidentally included SGD into the diffusive and bioturbative flux. Explaining the methods in full detail would be long and is not possible here, but we have shown the references.

Comment 7: Page 11, Line 20 and elsewhere: The comparison between SGD and river flow is appropriate and should be kept since it puts results in perspective. However, radiumderived SGD is likely to be saline water, while rivers are a source of fresh water. I encourage the authors to add a note qualifying those differences using the literature.

Answer 7: Page 11 line 20: Indeed, SGD is mainly recirculated saline water, as already explained in the introduction. According to Taniguchi [2007], there is only 2600 km$^3$/yr of fresh SGD, between 2% and 20% of the total SGD. The comparison with rivers is made because they both contain nutrients and trace elements. The end of 3.4 has been changed to explain all this more clearly, using the literature: "According to Taniguchi et al. (2007), there is only 2.6 x 10$^{12}$ m$^3$ yr$^{-1}$ of fresh groundwater discharge, representing between 2% and 20% of the SGD, and the remaining part is recirculated seawater."

Comment 8: Similar to the previous comment, the comparison to seepage meters on the last paragraph of page 12 may need to be qualified. Many seepage meter deployments are made in very shallow nearshore waters and may capture fresher SGD, while Ra-228 covers a much larger scale. Consider using radium studies to build this comparison.

Answer 8: Indeed, seepage meters are not representative of total SGD, as they are concentrated close to the coast, where both fresh and saline SGD are expected to be higher than the continental shelf average. This is why the fluxes of most local studies are one order of magnitude higher than ours. This is now explained more clearly by several small changes in 4.1. In the previous version of the manuscript, the only mentioned weakness of seepage meters was spatial variability, which was incomplete. However, seepage meters are interesting because they are independent from radium and are not affected by uncertainties on radium diffusion or radium concentration in groundwater. Not mentioning them would be strange. Radium studies, at a larger scale, are already the object of the previous paragraphs.

**Referee #2**

The manuscript was proofread again, and a few errors were corrected, including the use of numbers and abbreviations at the beginning of sentences.

Comment 1: I do not think the title accurately represents the main focus of the paper, which is an effort to improve the application of the inverse modeling technique to $^{228}$Ra, as opposed to an interpretation of an updated $^{228}$Ra flux estimate or SGD estimates. Either the title should be changed to better reflect the fact that the main contributions of this work are improvements to the model, or more discussion should be added on the interpretation of the results of the model and how they change our understanding of $^{228}$Ra and/or SGD. If the latter is chosen, the authors should be sure to highlight how their contributions are unique from those of Kwon et al (2014), aside from simply reporting the newly calculated Arctic and Southern Ocean $^{228}$Ra fluxes.

Answer 1: The title should reflect three aspects: a) We have shown that parameters such as the number of source regions and the cost function matter a lot in inverse modeling, b) By using appropriate parameters, we produce a more precise estimate than previous studies, c) However, this is not immediately usable for SGD studies as this source still have to be separated from diffusion. The current title might not emphasize the first aspect enough but finding a better one is difficult. The title has been changed to "Improving the inverse modeling of a trace isotope: how precisely can radium- 228 fluxes toward the ocean and Submarine Groundwater Discharge be estimated?"

Comment 2: There is a short discussion about the sources of Ra, and it is mentioned that dust inputs are small compared to the other sources used in the model. The dissolved riverine source should also be introduced here instead of later in the paper, because this is where the sources/sinks are first introduced. You state later on that the dissolved component of the riverine flux is negligible, but it would be better to approximate the relative contribution as you have done for the dust source in line 20.

Answer 2: Riverine dissolved radium-228 is less than 1% of the total radium-228 flux. This figure, similar to that of dust, has been added to the introduction (Page 2 line 23). More explanations are given at page 10, using the annual river discharge and the average concentration in dissolved radium: "According to Moore et al. (2008), the average dissolved $^{228}$Ra activity in rivers is 0.65-1.95 x $10^5$ atoms.l$^{-1}$. As rivers annually discharge 35000 km$^3$ of freshwater (Milliman, 2001), the dissolved riverine source lies between 2.3 and 6.8 x $10^{21}$ atoms.yr$^{-1}$, which is less than 1% of the total flux."

Comment 3: Page 7 lines 17 – 18: Why is the flux of $^{228}$Ra (in units of atoms y$^{-1}$) compared to river discharge (I'm assuming this is units of m$^3$ y$^{-1}$)? Why not compare the flux of SGD to river discharge, so that it is a volume-to-volume comparison? I'm not sure if this is intended to make the point that the largest flux of $^{228}$Ra does not correlate with the greatest river discharge, but the

authors argue that rivers do not carry much dissolved $^{228}$Ra, so in that case they shouldn't correlate anyway.

Answer 3: It would indeed make more sense to compare river discharge to the other large fluxes of water and nutrients to the ocean, Submarine Groundwater Discharge (SGD), rather than to radium-228 fluxes. The problem is that SGD suffers from high uncertainty, as highlighted in the article. Average basin-wide SGD fluxes can be compared, but this is implicitly assuming that diffusion per unit of surface is nearly the same in both basins, which is far from certain. This issue will be moved to the end of 3.4, where we now compare the distribution of river discharge with both radium fluxes and SGD, and explain why the latter is imprecise, although more interesting. The ratios are very close anyway. Page 11 lines 30-35 now read as: "In total, 63% of the $^{228}$Ra flows into the Indian and Pacific basins, which is more than their share of the continental shelf (52%). As shown in Fig.8, the estimated contributions of SGD to $^{228}$Ra fluxes are in very similar proportions, with 68% going to the Indian and Pacific basins. This represents more than 70% of all the groundwater discharge when the differences in groundwater $^{228}$Ra concentration are considered. However, uncertainties on SGD distribution are high, because diffusion from sediment might not be identical in all basins. These results contrast with the other source of water, nutrients and trace elements to the ocean: 60% of the global river discharge flows to the Atlantic and Arctic basins (Milliman, 2001)."

Comments 4 and 5: Indeed, I had called the wrong figures. This is now corrected

Comment 6: Conclusion lines 14 – 15: be specific about the other cost functions that were tested, instead of saying ". . .the other cost functions. . .". That way if a reader skips to the conclusion, they will still understand your specific results. The same sentence ends by saying ". . .more realistic assumptions on error statistics"; again, be specific about what the more realistic assumptions were. A large part of the paper is dedicated to testing assumptions, so the conclusion should point out the results of those tests.

Answer 6: The difference between the three cost functions and between the underlying assumptions on error statistics is now described explicitly in the conclusion, so that a reader skipping to the conclusion can understand what it is about and why it is important. Page 16 lines 2-6 now read as: "These precise estimates are obtained by minimizing the squared differences between model and observed concentrations on a logarithmic scale. We think this cost function is more realistic than a linear least-squares one because it assumes that the error standard deviation is proportional to $^{228}$Ra concentration, rather than constant, which proves to 5 be more accurate. It is also better than a "proportional" cost function, weighted by the inverse of the observed concentration, which tends to produce underestimations of concentrations and fluxes."

Comment 7: Conclusion lines 24 – 25: The sentence that begins with "Therefore, besides estimating the sources. . ." should not be included in the conclusion, as it is not one of the main

points emphasized in the paper. It is more appropriate to move this line to the end of the "model biases" section or another part of the manuscript.

Answer 7: This sentence will be removed from the conclusion, as it is not a main focus of the paper. This possibility of improving a circulation model using the residuals of inversions is already mentioned in 3.2 and 4.2 and there is not much to add.

Comment 8: The conclusion should have a stronger ending and remind the reader why it is important to better resolve the $^{228}$Ra model. Line 17 is a good reminder of why determining the shelf fluxes is important, but the conclusion should also highlight why it is important to separate the SGD and diffusion sources. This could also be discussed in more detail earlier in the paper, but should at least be mentioned again in the conclusion.

Answer 8: Evaluating the SGD is the goal of several studies and radium-228 has often been used as a tracer for this purpose. One important conclusion of this article is that using radium-228 in an inventory or in inverse model is very imprecise because SGD and diffusion are poorly separated. This is now stated more explicitly by changes in 3.4 (page 11 lines 18-29: "As long as diffusion cannot be separated from SGD more efficiently and is not shown to be negligible, noprecise SGD estimate based on $^{228}$Ra can be produced.") and in the conclusion ("These results mean that all the studies which have used or will use $^{228}$Ra to estimate SGD will suffer from a very high uncertainty until diffusion is properly estimated or proven to be negligible.").

Comment 9: Page 16 line 2: include the actual NEMO website

Answer 9: Done (page 16 line 23)

Comment 10: Full references for the data sources used in the model are not included in the reference list; is this because they are only listed in the supplementary material? I suggest adding a supplementary reference list for these data sources. Also, in the Arctic references, the year for the Rutgers van der Loeff et al. 2013 reference should be 2012, not 2013.

Answer 10: All the references used in our radium-228 dataset have been included in the supplement. The Rutgers van der Loeff et al. 2012 and two other mistakes have been corrected.

Comment 11: On Figure 9, the cruise across the Fram Strait has been completed, so this should be in yellow instead of red.

Answer 11: Corrected

[revised manuscript text omitted]
}\boldsymbol{x} - \mathbf{A_{inv}}\boldsymbol{b})(\mathbf{A_{inv}}\mathbf{A}\boldsymbol{x} - \mathbf{A_{inv}}\boldsymbol{b})^\top\rangle \\
&= \mathbf{A_{inv}}\langle(\mathbf{A}\boldsymbol{x} - \boldsymbol{b})(\mathbf{A}\boldsymbol{x} - \boldsymbol{b})^\top\rangle\mathbf{A_{inv}^\top} \\
&= \sigma^2\mathbf{A_{inv}}\mathbf{A_{inv}^\top}
\end{aligned} \tag{A3}$$

Equation (A3) corresponds to (2.102) in Wunsch (2006).

$\sigma^2$ is related to the root mean square of residuals $v_\eta$ the following way:

$$
\begin{aligned}
v_\eta &= \langle (\mathbf{A}\boldsymbol{x_{opt}} - \boldsymbol{b})^\top (\mathbf{A}\boldsymbol{x_{opt}} - \boldsymbol{b}) \rangle \\
&= \langle (\mathbf{H}\boldsymbol{b} - \boldsymbol{b})^\top (\mathbf{H}\boldsymbol{b} - \boldsymbol{
[revised manuscript text omitted]